# HIPK4 is essential for murine spermiogenesis

J Aaron Crapster[1]*, Paul G Rack[1], Zane J Hellmann[1], Austen D Le[1], Christopher M Adams[2], Ryan D Leib[2], Joshua E Elias[3], John Perrino[4], Barry Behr[5], Yanfeng Li[6], Jennifer Lin[6], Hong Zeng[6], James K Chen[1,7,8]*

[1]Department of Chemical and Systems Biology, Stanford University School of Medicine, Stanford, United States; [2]Stanford University Mass Spectrometry, Stanford University, Stanford, United States; [3]Chan Zuckerberg Biohub, Stanford University, Stanford, United States; [4]Cell Science Imaging Facility, Stanford University School of Medicine, Stanford, United States; [5]Department of Obstetrics and Gynecology, Reproductive Endocrinology and Infertility, Stanford University School of Medicine, Stanford, United States; [6]Transgenic, Knockout, and Tumor Model Center, Stanford University School of Medicine, Stanford, United States; [7]Department of Developmental Biology, Stanford University School of Medicine, Stanford, United States; [8]Department of Chemistry, Stanford University, Stanford, United States

**Abstract** Mammalian spermiogenesis is a remarkable cellular transformation, during which round spermatids elongate into chromatin-condensed spermatozoa. The signaling pathways that coordinate this process are not well understood, and we demonstrate here that homeodomain-interacting protein kinase 4 (HIPK4) is essential for spermiogenesis and male fertility in mice. HIPK4 is predominantly expressed in round and early elongating spermatids, and *Hipk4* knockout males are sterile, exhibiting phenotypes consistent with oligoasthenoteratozoospermia. *Hipk4* mutant sperm have reduced oocyte binding and are incompetent for in vitro fertilization, but they can still produce viable offspring via intracytoplasmic sperm injection. Optical and electron microscopy of HIPK4-null male germ cells reveals defects in the filamentous actin (F-actin)-scaffolded acroplaxome during spermatid elongation and abnormal head morphologies in mature spermatozoa. We further observe that HIPK4 overexpression induces branched F-actin structures in cultured fibroblasts and that HIPK4 deficiency alters the subcellular distribution of an F-actin capping protein in the testis, supporting a role for this kinase in cytoskeleton remodeling. Our findings establish HIPK4 as an essential regulator of sperm head shaping and potential target for male contraception.

*For correspondence:
aaron.crapster@vibliome.com (JAC);
james.chen@stanford.edu (JKC)

## Introduction

Spermiogenesis is a critical, post-meiotic phase of male gametogenesis defined by the differentiation of spermatids into spermatozoa (*Figure 1A–B*; *Russell et al., 1990*). This dramatic morphological transformation is mediated by a series of cytological processes that are unique to the testis (*Kierszenbaum et al., 2007*; *O'Donnell, 2014*). In round spermatids, Golgi-derived vesicles give rise to the acrosome (*Berruti and Paiardi, 2011*), a cap-like structure that is anchored to the anterior nuclear membrane by a filamentous actin (F-actin)- and keratin 5-containing plate called the acroplaxome (*Kierszenbaum et al., 2003a*; *Kierszenbaum et al., 2004*). Neighboring Sertoli cells form an apical specialization that circumscribes each spermatid head, and F-actin hoops within these anchoring junctions apply external forces to the acrosome–acroplaxome complex and underlying spermatid nucleus (*Wong et al., 2008*). The posterior nuclear pole in spermatids is simultaneously girdled by a

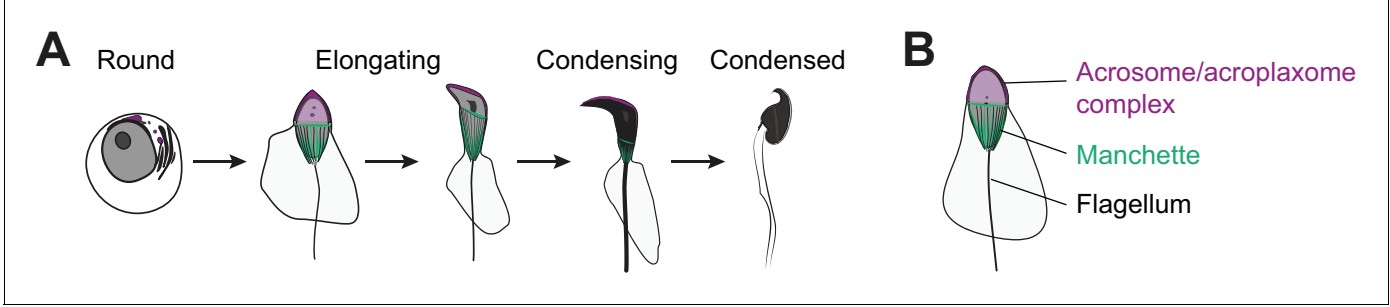

**Figure 1.** Key steps of spermiogenesis. (A) Schematic representation of murine male germ cells transitioning from round spermatids to elongated spermatozoa. These steps occur within the testis seminiferous epithelium and are conserved in all mammals. (B) Illustration of an elongating spermatid highlighting structural features that are established during spermiogenesis.

transient microtubule- and F-actin-scaffolded structure called the manchette, which extends from the basal body of the developing flagellum and is separated from the acrosome–acroplaxome complex by a narrow groove (*Kierszenbaum and Tres, 2004*; *Lehti and Sironen, 2016*).

As spermiogenesis proceeds, these membranous and cytoskeletal structures act in concert to elongate the spermatid head. The spermatid nucleus becomes highly compact as chromatin condenses into a quiescent state (*Rathke et al., 2014*), and the germ cell expels its cytoplasmic contents through residual bodies and actin-scaffolded tubulobulbar complexes that contact neighboring Sertoli cells (*Sprando and Russell, 1987*; *Upadhyay et al., 2012*; *Zheng et al., 2007*). The acrosome–acroplaxome complex and manchette concurrently mediate protein transport from the Golgi to the developing flagellum, delivering cargoes required for flagellum assembly and function (*Kierszenbaum et al., 2011*; *Kierszenbaum et al., 2004*). In mature sperm, the acrosome then promotes sperm-egg fusion through the exocytotic release of digestive enzymes and the display of oocyte-binding receptors that are localized to the inner acrosomal membrane (*Stival et al., 2016*).

In contrast to these detailed cytological descriptions, our understanding of the molecular mechanisms that coordinate spermiogenesis is still nascent. Initial insights into this process have been provided by mouse mutants with spermatogenic and male fertility defects (*de Boer et al., 2015*; *Yan, 2009*). For example, pro-acrosomal vesicles fail to fuse in mice that lack the nucleoporin-like protein HRB/AGFG1 (*Kang-Decker et al., 2001*; *Kierszenbaum et al., 2004*), nuclear membrane protein DPY19L2 (*Pierre et al., 2012*), certain Golgi-associated proteins [GOPC (*Yao et al., 2002*), PICK1 (*Xiao et al., 2009*), and GM130 (*Han et al., 2018*), or acrosomal factors [SPACA1 (*Fujihara et al., 2012*) and SPATA16 (*Fujihara et al., 2017*)]. These mutants consequently produce acrosome-less sperm with rounded heads—defects that are characteristic of globozoospermia. Acrosome biogenesis also requires the matrix component ACRBP (*Kanemori et al., 2016*) and the coiled coil protein CCDC62 (*Li et al., 2017*; *Pasek et al., 2016*), and loss of either acrosomal protein can cause phenotypes resembling oligoasthenoteratozoospermia (OAT), a fertility disorder characterized by low sperm concentrations and spermatozoa with abnormal shapes and reduced motility (*Tüttelmann et al., 2018*).

Murine models have similarly revealed proteins that are required for manchette and flagellum assembly, including the RIMBP3-HOOK1 (*Zhou et al., 2009*), LRGUK1-HOOK2 (*Liu et al., 2015*), MEIG1-PACRG-SPAG16L (*Li et al., 2015*), and FU (*Nozawa et al., 2014*). Manchette shaping and degradation are also essential for sperm development, as demonstrated by the OAT-like phenotypes of mice expressing a loss-of-function variant of the microtubule-severing protein Katanin p80 (*O'Donnell et al., 2012*). As spermiogenesis proceeds, membranous and cytoskeletal structures are dynamically coupled by distinct LINC (Linker of Nucleoskeleton and Cytoskeleton) complexes. These include LINC components that reside in the outer acrosomal membrane (SUN1 and nesprin3) (*Göb et al., 2010*) or posterior nuclear envelope (SUN3, SUN4, SUN5, and nesprin1) (*Göb et al., 2010*; *Pasch et al., 2015*; *Shang et al., 2017*). For example, loss of SUN4 function in mice causes manchette disorganization, sperm head defects, and male sterility.

Factors that specifically contribute to acroplaxome function have been more difficult to identify and study. Actin-binding proteins such as myosins Va and VI, profilins III and IV, and cortactin localize

to the acroplaxome and have been implicated in its regulation (*Behnen et al., 2009*; *Kierszenbaum et al., 2003b*; *Kierszenbaum et al., 2008*; *Kierszenbaum et al., 2011*; *Zakrzewski et al., 2017*); however, their wide-spread expression in somatic tissues has hindered functional studies. One notable exception is the actin-capping protein, CAPZA3, a spermatid-specific factor that associates with CAPZB3 and binds to the barbed ends of F-actin. *Capza3* mutant male mice are sterile and have OAT-like defects, indicating that F-actin dynamics within the acroplaxome play an important role in spermiogenesis (*Geyer et al., 2009*).

Upstream signaling proteins that control cytoskeletal dynamics are likely to be critical drivers of spermatid differentiation. For instance, PLCγ−1 phosphorylation is dysregulated in the germ cells of KIT[D814Y] mutant mice, leading to mislocalized manchettes and deformed spermatid heads (*Schnabel et al., 2005*). Phosphoproteomic analyses indicate that several kinase-dependent pathways are active throughout sperm development, but the roles of specific kinases in spermiogenesis are not well understood (*Castillo et al., 2019*). Here, we describe an essential function for homeodomain-interacting protein kinase 4 (HIPK4) in murine spermiogenesis and fertility. This dual-specificity kinase is predominantly expressed in the testis, where it is restricted to round and early elongating spermatids. Male *Hipk4* knockout mice are sterile and exhibit spermatogenic defects characteristic of OAT. Sperm produced by these mutant mice are also incompetent for oocyte binding and in vitro fertilization, and they exhibit head defects associated with dysregulation of the acrosome–acroplaxome complex. Consistent with these observations, HIPK4 overexpression in cultured somatic cells remodels the F-actin cytoskeleton and alters the phosphorylation state of multiple actin-interacting proteins. In the testis, HIPK4 co-fractionates with F-actin and HIPK4 deficiency alters cytoskeletal interactions with an F-actin capping protein. Taken together, our studies demonstrate that HIPK4 regulates the actin cytoskeleton, acrosome–acroplaxome dynamics, spermatid head shaping, and ultimately, sperm function.

## Results

### HIPK4 is expressed in differentiating spermatids

Gene expression data available through the Genotype Tissue Expression Project (https://www.gtex-portal.org) and the Mammalian Reproductive Genetics Database (http://mrgd.org) indicate that HIPK4 is largely expressed in the testis, with lower levels detected in the brain. Using a tissue cDNA array and quantitative PCR, we also found that *Hipk4* is robustly transcribed in the adult murine testis (*Figure 2A*). In situ hybridization of testis sections obtained from 8-week-old C57BL/6NJ mice revealed that *Hipk4* is transcribed specifically in round and early elongating spermatids (*Figure 2B*), and we observed comparable *HIPK4* expression patterns in adult human testis samples (*Figure 2C*). We then assayed testis sections from juvenile male mice of different ages to determine precisely when *Hipk4* is expressed during spermatogenesis, taking advantage of the initial, synchronized wave of male germ cell development. *Hipk4* transcripts were first detected in germ cells at 21 days postpartum (dpp), coinciding with the appearance of step 2–3 round spermatids (*Figure 2—figure supplement 1*). The population of *Hipk4*-positive spermatids expanded until 29 dpp, at which point *Hipk4* mRNA became undetectable in elongating spermatids circumscribing the seminiferous lumen. These results suggest that HIPK4 specifically functions within male germ cells as they transition from round to elongating spermatids.

### HIPK4 is essential for male fertility

*Hipk4* knockout mice were first reported in a 2008 patent application by Bayer Schering Pharma, which described general defects in sperm morphology and number (*Sacher et al., 2008*). However, the fertility of these mutant mice was not characterized, nor were the mice made publicly available. As part of the Knockout Mouse Phenotyping Program (KOMP[2]), the Jackson Laboratory generated mice containing a *Hipk4* null allele (*tm1b*), in which a β-galactosidase reporter replaces exons 2 and 3. We established a colony of *Hipk4*[tm1b/tm1b] mice (henceforth referred to as *Hipk4*[−/−]) and confirmed that these mice fail to produce functional *Hipk4* gene products using genomic PCR and western blot analysis (*Figure 2D*). Loss of HIPK4 had no apparent effect on the animal viability or growth (*Figure 2E*). By immunostaining testis cryosections from adult wild-type and *Hipk4*[−/−] mice, we confirmed that HIPK4 protein is expressed in round and early elongating spermatids (steps 3–8). The

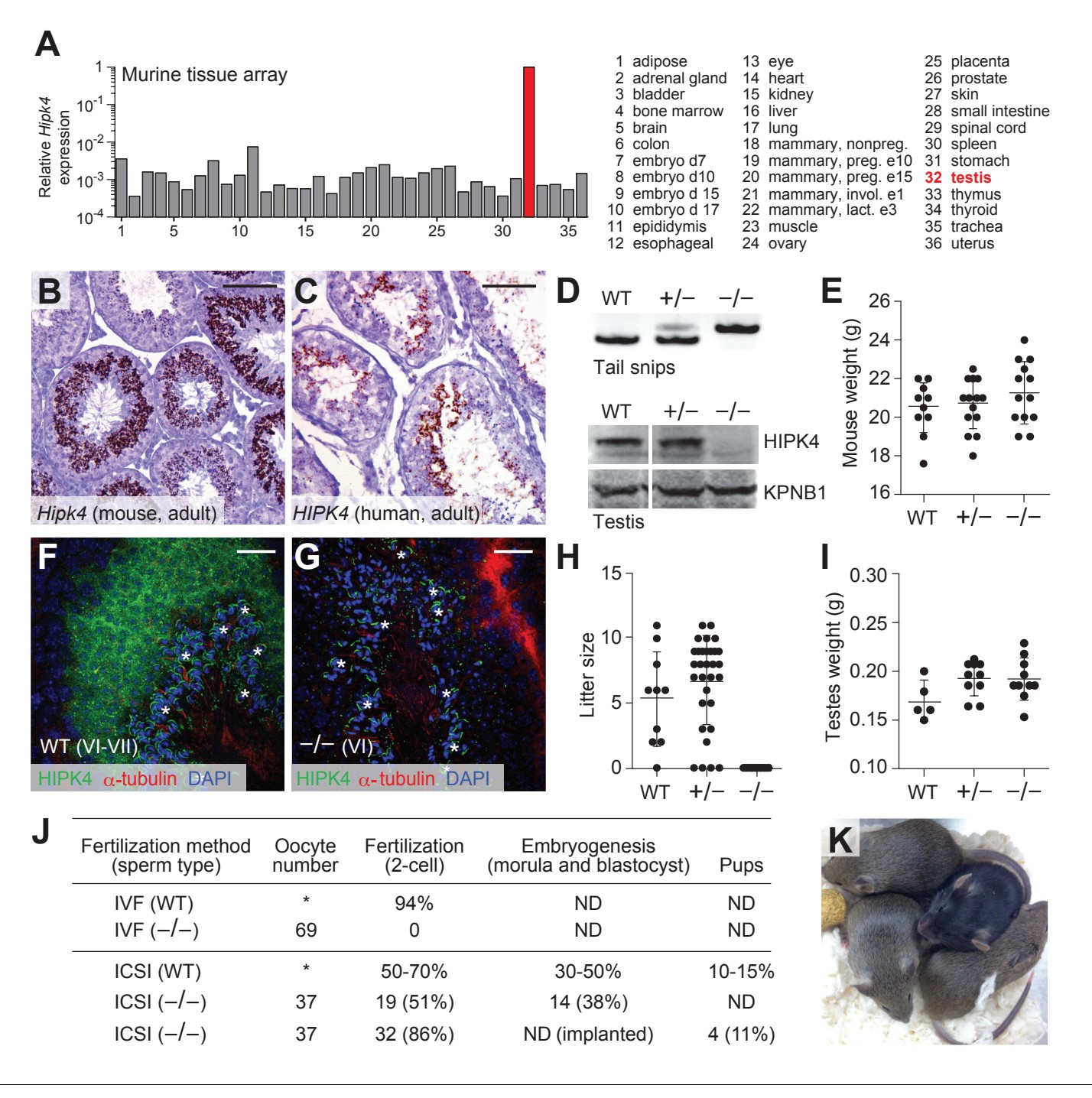

**Figure 2.** HIPK4 is expressed in spermatids and required for male fertility in mice. (**A**) *Hipk4* expression in various murine tissues as determined by qPCR analysis of the Origene TissueScan Mouse Normal cDNA array. Data are normalized to *Gapdh*. (**B–C**) *Hipk4* expression in adult mouse (**B**) and human (**C**) testis sections as determined by in situ hybridization. (**D**) Validation of *Hipk4* knockout by PCR of tail-derived genomic DNA and western blot analyses of whole testis lysates. Immunoblots are from the same membrane and exposure time. (**E**) Weights of WT, *Hipk4$^{+/-}$*, and *Hipk4$^{-/-}$* males at 6–7 weeks of age. (**F–G**) HIPK4 protein expression in adult mouse seminiferous tubule sections (Stage VI-VII) as determined by immunofluorescence imaging. Asterisks indicate non-specific antibody binding (see *Figure 2—figure supplement 2*). (**H**) Number of live pups per litter resulting from crosses between 7-week-old males and age-matched, WT females. (**I**) WT, *Hipk4$^{+/-}$*, and *Hipk4$^{-/-}$* testis weights at 6 weeks of age. (**J**) Fertilization potential of *Hipk4$^{-/-}$* sperm using IVF and ICSI. ND = not determined. Experiments were conducted by the Stanford Transgenic, Knockout, and Tumor Model Center, and the wild-type data represent the core facility's average results using C57BL/6NJ sperm. (**K**) Pups born via ICSI using *Hipk4$^{-/-}$* sperm. Scale bars: B-C, 100 µm; F-G, 20 µm. Statistical analyses: error bars depicted in panels E, H, and I represent the average value ± s.d.

*Figure 2 continued on next page*

*Figure 2 continued*

The online version of this article includes the following figure supplement(s) for figure 2:

**Figure supplement 1.** *Hipk4* mRNA expression during the first wave of murine spermatogenesis.

**Figure supplement 2.** HIPK4 protein expression in adult murine seminiferous tubules.

kinase is distributed throughout the cytoplasm of these germ cells, mirroring its subcellular localization when overexpressed in somatic cells (*Figure 2F–G*, *Figure 2—figure supplement 2*; *van der Laden et al., 2015*).

Despite their grossly normal development and physiology, homozygous *Hipk4* mutant males were unable to conceive, whereas heterozygote males sired normal litter sizes (*Figure 2H*). No significant differences in testis weight were observed across the *Hipk4* genotypes (*Figure 2I*). Female *Hipk4*$^{-/-}$ mice were physically indistinguishable from their wild-type and heterozygous littermates and gave birth to normal litter sizes (data not shown). Epididymal sperm isolated from *Hipk4*$^{-/-}$ mice failed to fertilize wild-type oocytes under standard in vitro fertilization (IVF) conditions, but intracytoplasmic sperm injection (ICSI) of the mutant sperm yielded embryos that could undergo uterine implantation to produce healthy pups (*Figure 2J–K*).

## HIPK4-deficient mice exhibit OAT-like phenotypes

During our in vitro fertilization studies, it was apparent that the *Hipk4*$^{-/-}$ male mice had spermatogenesis defects consistent with OAT. In comparison to wild-type mice, homozygous mutants produced sperm at low epididymal concentrations, and the germ cells had decreased motility [both the total number of motile sperm and those with progressive motility as measured by computer-assisted sperm analysis (CASA)] and abnormal morphology (*Figure 3A–E*, *Figure 3—figure supplement 1*). Head defects included macrocephaly, microcephaly, and irregular shapes; tail deformities included bent, coiled, crinkled, and shortened flagella. *Hipk4*$^{+/-}$ sperm also had reduced epididymal concentrations and total motility, but their progressive motility and morphology were normal. We further observed that over 10% of the epididymal sperm isolated from *Hipk4*$^{-/-}$ mice exhibited DNA fragmentation that could be detected by TUNEL (terminal deoxynucleotidyl transferase dUTP nick end labeling) staining, whereas only 1.5% of wild-type or *Hipk4*$^{+/-}$ sperm were TUNEL-positive (*Figure 3F*). We found no evidence of of increased TUNEL staining in the testes of mutant mice (data not shown). Homozygous mutant sperm therefore may undergo a higher rate of apoptosis after spermiation, possibly accounting for their lower epididymal concentrations.

We further compared the head structures of wild-type and *Hipk4*$^{-/-}$ sperm by scanning electron microscopy (SEM; *Figure 3G*). All HIPK4-deficient sperm exhibited head morphologies that deviated from the flat, crescent-shaped structures of their wild-type counterparts. Specific defects included a disorganized anterior acrosome and the absence of a distinct equatorial segment, post-acrosomal sheath, ventral spur, and sharp hook rim. In some sperm samples with demembranated head structures, we observed mislocalized axonemal components wrapped around the nucleus. We also used transmission electron microscopy (TEM) to analyze the tail structures of newly formed spermatozoa within the seminiferous tubule. HIPK4 loss did not appear to affect the basal body (*Figure 3H–H'*), axoneme (*Figure 3I–K'*), mitochondria (*Figure 3I–I'*), outer dense fibers (*Figure 3J–J'*), or fibrous sheath (*Figure 3K–K'*) of these fully differentiated cells.

## HIPK4-deficient sperm can undergo capacitation and the acrosome reaction in vitro

We next examined how the loss of HIPK4 affects two key aspects of sperm function: capacitation and the acrosome reaction. Sperm naturally undergo capacitation as they ascend the female reproductive tract and interact with the oviduct epithelium (*Austin, 1951*; *Chang, 1951*). Changes in the glycoproteins, phospholipids, and cholesterol residing in the sperm plasma membrane, Ca$^{2+}$ influx, and increases in intracellular pH correlate with the activation of soluble adenylate cyclase and protein kinase A (*Abou-haila and Tulsiani, 2009*; *Lin and Kan, 1996*; *O'Rand, 1982*; *Stival et al., 2016*). Downstream protein tyrosine phosphorylation signaling events are initiated, and sperm switch from progressive motility to a 'hyperactivated' swimming motion (*Alvau et al., 2016*; *Naz and Rajesh, 2004*; *Sepideh et al., 2009*). Capacitated sperm also become competent for the

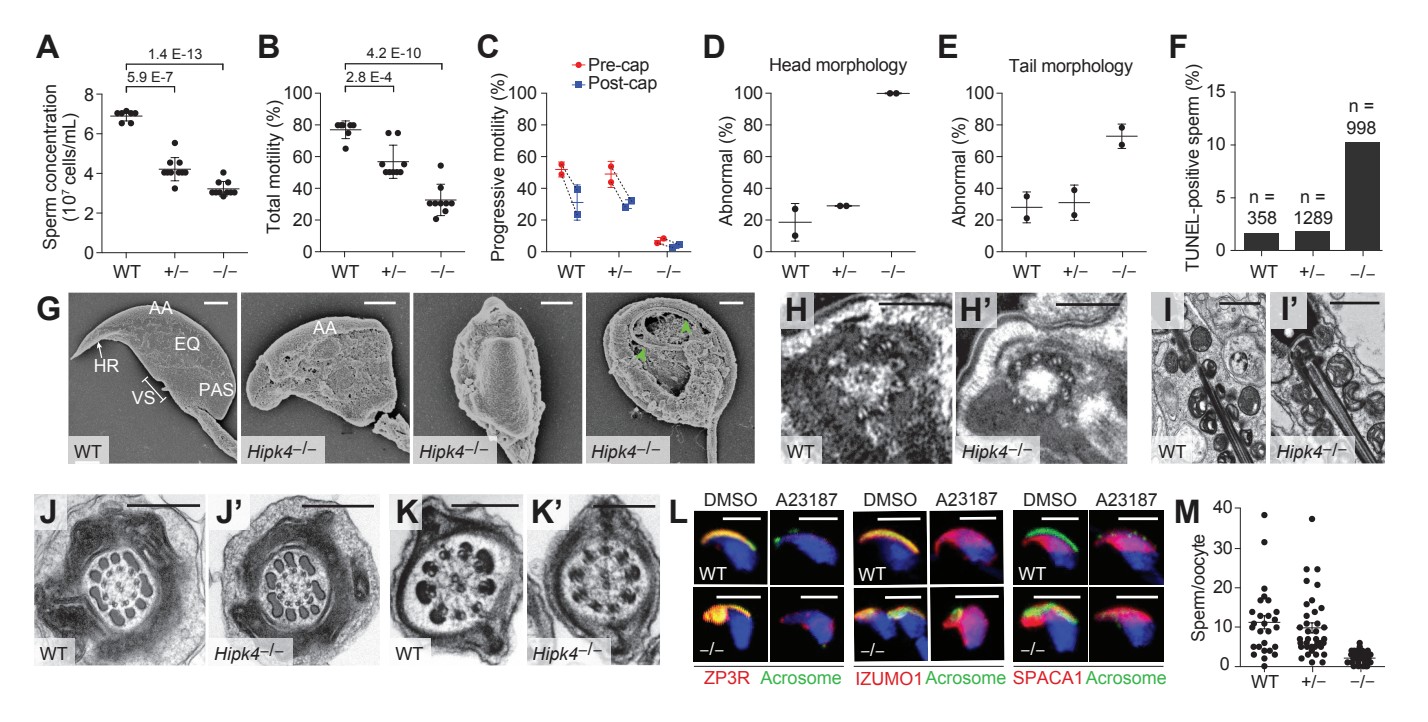

**Figure 3.** HIPK4 knockout mice exhibit oligoasthenoteratozoospermia. (**A**) Concentrations of epididymal sperm ± s.d. P values for the indicated statistical comparison are shown. (**B**) Percentage of epididymal sperm that were motile ± s.d. P values for the indicated statistical comparison are shown. (**C**) CASA measurements of progressive sperm motility. Data connected by the dashed lines are showing motility of >500 sperm from the same samples before and after capacitation ± s.d. (**D, E**) Percentage of sperm with abnormal head or tail morphology ± s.d. as assessed by phase contrast microscopy (>500 sperm analyzed for each genotype; see also *Figure 3—figure supplement 1* for phase contract images). (**F**) Quantification of TUNEL-positive epididymal sperm from an individual male of each genotype (n = number of sperm analyzed). (**G**) SEM images of epididymal sperm. AA = anterior acrosome, EQ = equatorial segment, PAS = postacrosomal segment, VS = ventral spur, HR = hook rim. Arrowheads point to the axoneme wrapped inside a demembranated sperm head. (**H, H'**) Centrioles at the basal body of step 15 spermatid axonemes. (**I, I'**) Mitochondria along the midpiece of step 15 spermatids. (**J, J'**) Cross section of step 15 spermatid flagella at the midpiece. (**K, K'**) Cross section of step 15 spermatid flagella at the principal piece. (**L**) Acrosomal changes in capacitated sperm treated with a Ca²⁺ ionophore (A23187) as assessed by immunofluorescence and staining with FITC-labeled PNA. Nuclei were stained with DAPI. (**M**) Number of oocyte-bound sperm under standard IVF conditions after extensive washing ± s.e.m. Data are from a single experiment using 24–30 COCs for each sperm incubation. Scale bars: G, 2 µm; H, 1 µm; I, 1 µm; J, 0.5 µm; K, 0.2 µm; L, 5 µm.

The online version of this article includes the following figure supplement(s) for figure 3:

**Figure supplement 1.** Morphologies of wild-type and *Hipk4* mutant sperm.

**Figure supplement 2.** HIPK4 is not essential for sperm capacitation or acrosomal exocytosis.

acrosome reaction, during which the outer acrosomal membrane fuses with the overlying plasma membrane (*Hirohashi, 2016*; *Kierszenbaum, 2000*; *Stival et al., 2016*). This process causes the release of digestive enzymes stored within the acrosome, and it exposes oocyte-binding receptors that are displayed on the inner acrosomal membrane, which spreads down over the equatorial region of the sperm head (*Sebkova et al., 2014*; *Sosnik et al., 2009*). Together, these steps ultimately promote sperm-oocyte engagement, fusion, and fertilization.

To determine whether HIPK4 is required for sperm capacitation, we isolated motile sperm from the caudal epididymides of wild-type, *Hipk4⁺/⁻*, and *Hipk4⁻/⁻* mice and incubated them in capacitation medium containing Ca²⁺, bicarbonate, and bovine serum albumin. We then assessed the resulting levels of tyrosine phosphorylation (p-Tyr) in soluble sperm lysates by western blot. We observed no significant differences in p-Tyr between the three *Hipk4* genotypes (*Figure 3—figure supplement 2A*), and by CASA, we found that wild-type, *Hipk4⁺/⁻*, and *Hipk4⁻/⁻* sperm treated with capacitation medium undergo similar changes in progressive motility (*Figure 3C*). These results indicate that *Hipk4⁻/⁻* sperm are competent for capacitation in vitro.

To investigate the ability of HIPK4-deficient sperm to undergo acrosomal exocytosis, we incubated capacitated sperm with the $Ca^{2+}$ ionophore A23187. Sperm were then fixed, quickly permeabilized, and labeled with fluorescein-conjugated peanut agglutinin (FITC-PNA), which binds to the outer acrosome membrane and is lost upon exocytosis. After 1.5 hr of capacitation, 86% of wild-type sperm retained fully intact acrosomes, whereas only 63% of mutant sperm had acrosomal FITC-PNA staining (due to either their severe head malformations or spontaneous acrosome exocytosis) (*Figure 3—figure supplement 2B*). As expected, A23187 treatment decreased the percentage of FITC-PNA-positive cells in both wild-type and *Hipk4*$^{-/-}$ samples, although mutant sperm responded less efficiently to this treatment (74% and 29% reductions in FITC-PNA-positive cells, respectively).

Finally, we examined specific acrosomal proteins during A23187-induced exocytosis: ZP3R, IZUMO1, and SPACA1. ZP3R (also known as sp56) is a zona pellucida-binding protein that appears on the sperm surface after capacitation, and it is released when the outer acrosomal and plasma membranes fuse (*Kim et al., 2001*). IZUMO1 (*Inoue et al., 2005*; *Sebkova et al., 2014*) and SPACA1 (*Fujihara et al., 2012*) are membrane proteins required for head shaping and oocyte fusion that localize to distinct acrosomal regions. IZUMO1 spreads from the anterior acrosomal cap to the equatorial segment during the acrosome reaction (*Inoue et al., 2005*; *Sebkova et al., 2014*), and SPACA1 remains localized to the equatorial region (*Fujihara et al., 2012*). As assessed by immunofluorescence microscopy, all three acrosomal factors exhibited HIPK4-independent behaviors in response to $Ca^{2+}$ ionophore exposure (*Figure 3L*). Although HIPK4-deficient sperm are capable of normal acrosomal changes, at least in response to calcium signaling, our data support a role for HIPK4 in promoting normal acrosome structure and exocytosis.

## HIPK4-deficient sperm exhibit diminished oocyte binding

Since *Hipk4*$^{-/-}$ sperm retain their ability to undergo capacitation and the acrosome reaction in vitro, we considered whether the head defects caused by HIPK4 loss might compromise sperm-oocyte interactions. Equivalent numbers of motile, capacitated sperm from wild-type, *Hipk4*$^{+/-}$, or *Hipk4*$^{-/-}$ males were incubated with cumulus-oocyte complexes (COCs) in human tubal fluid (HTF) supplemented with $Ca^{2+}$ and glutathione. The complexes were then washed repeatedly, fixed, and oocyte-bound sperm were quantified by nuclear staining and confocal imaging (*Figure 3M*). Consistent with the incompetence of *Hipk4*$^{-/-}$ sperm for IVF, these cells bound less efficiently to oocytes in comparison to their wild-type and heterozygous mutant counterparts. We further noted that the COCs incubated with mutant sperm retained many cumulus cells, while all of the cumulus cells of the COCs exposed to wild-type sperm were detached (*Figure 3—figure supplement 2C*), as expected from the hyaluronidase and hexosaminidase activities of normal sperm (*Lin et al., 1994*; *Zao et al., 1985*). Thus, HIPK4-deficient sperm lack structural and/or molecular features that are required for maximally productive sperm-oocyte interactions.

## Loss of HIPK4 function disrupts acrosome–acroplaxome interactions

To gain insights into the spermatogenic defects caused by loss of HIPK4, we compared periodic acid-Schiff (PAS)-stained testis sections obtained from adult wild-type and *Hipk4*$^{-/-}$ mice. *Hipk4*$^{-/-}$ testes contained malformed elongating spermatids, which failed to properly extend by step 12 (*Figure 4A–A'*, *Figure 4—figure supplement 1A–B*). Despite their morphological abnormalities, *Hipk4*$^{-/-}$ spermatozoa released to the epididymis progressed normally to the cauda (*Figure 4B–B'*, *Figure 4—figure supplement 1C–D*).

Next, we analyzed spermatid structures in greater detail by TEM imaging of testis sections. Although HIPK4 protein expression peaks in step 5–7 spermatids, all *Hipk4*$^{-/-}$ round spermatids appeared normal by TEM (*Figure 4C–C'*, *Figure 4—figure supplement 2A*). Spermatids begin to elongate at step 8, and most of these cells appeared normal in *Hipk4*$^{-/-}$ testes. However, some HIPK4-null step eight spermatids contained highly amorphous, fragmented acrosomal vesicles and/or detached acrosomal granules, which were coincident with structural abnormalities to the anterior nuclear pole (*Figure 4—figure supplement 2A*). Aberrant head structures became universally apparent in step 9–10 *Hipk4*$^{-/-}$ spermatids (*Figure 4C–C'*). The posterior edge of the acrosome was no longer juxtaposed to the perinuclear ring of the manchette, significantly widening the groove belt and deforming the underlying nuclear lamina (*Figure 4D–D'*). In some cases, enlargement of the groove belt was accompanied by detachment of the acrosome, severe anterior head deformities,

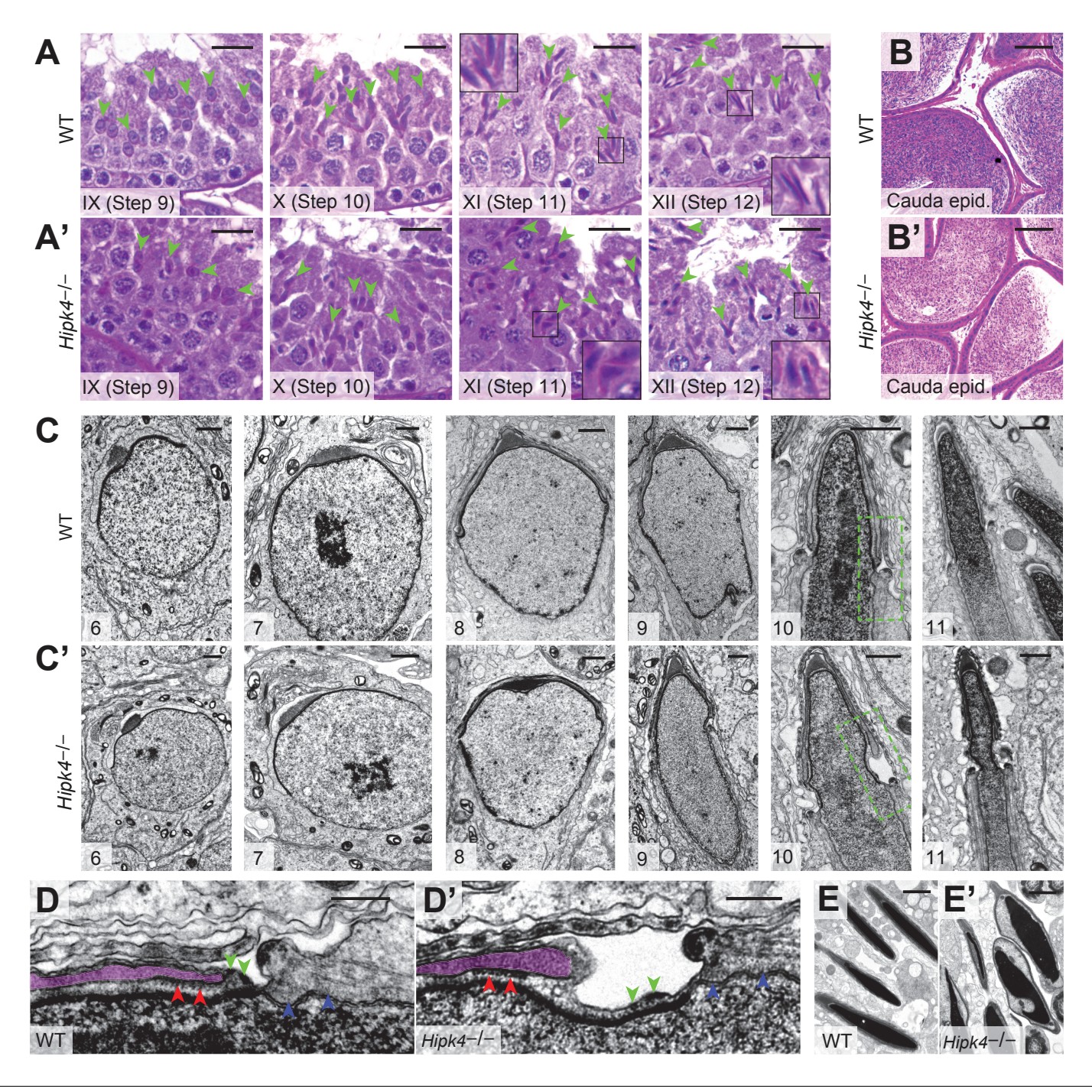

**Figure 4.** HIPK4 regulates the acrosome–acroplaxome complex. (A,A') PAS-stained sections of seminiferous tubules from WT or *Hipk4⁻/⁻* mice at stages IX-XII of spermatogenesis. Green arrowheads point to the heads of representative elongating spermatids (Step 9–12, respectively). (B, B') H and E-stained sections of caudal epididymides from WT or *Hipk4⁻/⁻* mice. (C–E) TEM images. (C, C') Step 6–11 spermatids from WT or *Hipk4⁻/⁻* mice. The areas outlined by green boxes are shown at higher magnification in D-D'. (D, D') Groove belt region of step 10 spermatids. Green arrowheads label to electron densities in the acroplaxome that are normally associated with the posterior edge of the acrosome (false-colored purple). Red arrowheads indicated keratin 5 filaments within the acroplaxome marginal ring, and blue arrowheads point to filaments linking the manchette to the nuclear envelope. (E, E') TEM images of condensed spermatids in WT and *Hipk4⁻/⁻* testis sections. Scale bars: A-B, 20 µm; C, 1 µm; D, 0.2 µm; E, 2 µm. The online version of this article includes the following figure supplement(s) for figure 4:

**Figure supplement 1.** Comparison of the seminiferous epithelium and epididymis of WT and *Hipk4* knockout mice.

**Figure supplement 2.** HIPK4 null spermatids exhibit acrosome–acroplaxome defects.

*Figure 4 continued on next page*

*Figure 4 continued*

**Figure supplement 3.** HIPK4 does not regulate the localization of anterior LINC complexes or manchette dynamics.

and retention of spermatid cytoplasm (*Figure 4—figure supplement 2B*). Keratin five filaments within the acroplaxome marginal ring appeared to be structurally intact and properly anchored to the acrosome in *Hipk4⁻/⁻* spermatids (red arrowheads in *Figure 4D–D'*). However, we observed small electron-dense areas at the posterior boundary of the acroplaxome that have not been previously defined. These electron-dense plaques adjoin the acrosome and nuclear lamina in wild-type but not HIPK4-null spermatids (green arrowheads in *Figure 4D–D'*).

Despite the acroplaxome-acrosome defects observed by electron microscopy, previously identified molecular components of these anterior head structures appeared to form properly in HIPK4-deficient spermatids by immunofluorescence imaging. For example, the anterior nuclear membrane (nuclear dense lamina) putatively anchors the acroplaxome *via* the inner membrane protein DPY19L2, while the outer acrosomal membrane is connected to the cytoskeleton via a LINC complex composed of SUN1 and nesprin3. These membrane-cytoskeleton linkages had comparable subcellular distributions in wild-type and *Hipk4⁻/⁻* spermatids, as assessed by immunofluorescence imaging of isolated germ cells (*Figure 4—figure supplement 3A–B*). TEM analyses similarly revealed no overt defects within the manchette, including the perinuclear ring and the conical array of filaments that scaffold the posterior nuclear pole (*Figure 4C–C' and D–D'*). We further characterized manchette formation and degradation by immunostaining the microtubule end-binding protein EB3 in testis cryosections and isolated germ cells. Testis sections from *Hipk4⁻/⁻* mice exhibited wild-type-like manchette dynamics (*Figure 4—figure supplement 3C*). However, elongating *Hipk4⁻/⁻* spermatids isolated from dissociated testis tissues occasionally exhibited abnormally elongated manchettes and narrow perinuclear rings (*Figure 4—figure supplement 3D*), suggesting that HIPK4 may also regulate certain aspects of this microtubule- and F-actin-scaffolded structure.

## HIPK4 does not primarily act through transcriptional regulation

Given the apparent delay between the onset of HIPK4 expression in wild-type round spermatids and the emergence of HIPK4-dependent morphological phenotypes, we investigated whether HIPK4 signaling could regulate the acrosome–acroplaxome complex through transcriptional mechanisms. Other HIPK family members and related kinases (*e.g.*, DYRKs) localize to the nucleus and have established roles in transcriptional regulation (*Di Vona et al., 2015*; *Rinaldo et al., 2008*), and HIPK4 can phosphorylate p53 in vitro (*Arai et al., 2007*). We therefore used oligonucleotide microarrays with full-genome coverage to profile transcriptional differences between adult testes isolated from wild-type and *Hipk4⁻/⁻* mice. Through this approach, we identified 415 genes that were upregulated ≥2 fold in *Hipk4⁻/⁻* testes compared to wild-type tissue and 709 genes that were downregulated ≥2 fold (*Figure 5A* and *Figure 5—source data 1*). Hierarchical clustering of the data from biological replicates revealed overlapping groupings of wild-type and *Hipk4⁻/⁻* samples (*Figure 5B*), indicating that the transcriptional differences between the genotypes are modest. *Hipk4* itself exhibited the largest change in transcript abundance between genotypes (*Figure 5C*). We also noted that loss of HIPK4 did not significantly alter the mRNA levels of transcription factors that are present in step 5–7 spermatids (*Green et al., 2018*). These results suggest that HIPK4 acts primarily through non-transcriptional mechanisms.

## HIPK4 overexpression remodels F-actin in cultured somatic cells

In the absence of discrete HIPK4-dependent transcriptional programs, we investigated the biochemical functions of HIPK4 in cells. We retrovirally overexpressed wild-type *Hipk4* or a catalytically dead mutant, *Hipk4^K40S^* (*Arai et al., 2007*), in cultured mouse embryonic fibroblasts (NIH-3T3 cells). Untransduced fibroblasts or those expressing the K40S mutant maintained spindle-like morphologies, whereas cells expressing HIPK4 became either spherical or polygonal within two days after infection. The polygonal HIPK4-overexpressing cells were multinucleate, likely due to cytokinesis failure (*Figure 6A–C*). Coincident with these changes in cell shape, we observed a striking loss of F-actin-containing stress fibers in the HIPK4-overexpressing cells (*Figure 6D–E*). As determined by ultracentrifugation of the corresponding cell lysates, the overall ratio of soluble, globular actin (G-

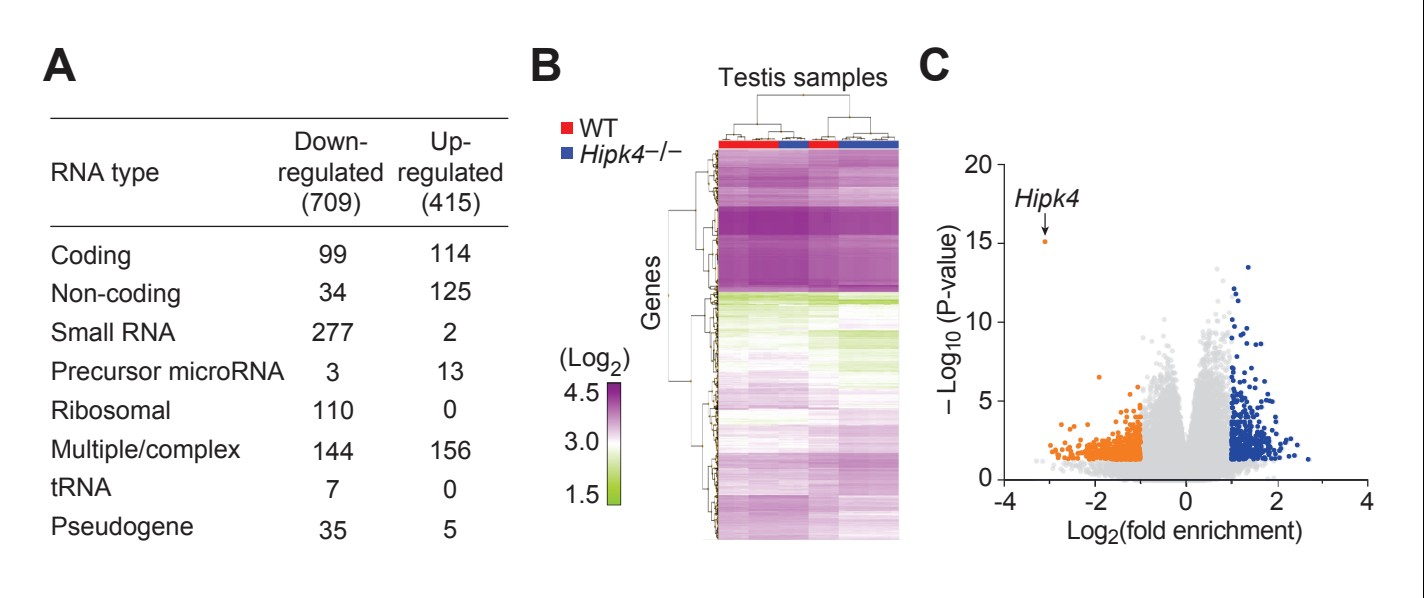

**Figure 5.** Wild-type and HIPK4 knockout testes have similar transcriptomes. (A) Summary of the types of RNA that were increased (up-regulated) or decreased (down-regulated) in *Hipk4* knockout testes compared to wild-type, as measured by quadruplicate microarray analysis of three testis samples. (B) Histogram of normalized signal intensities (Log2) of gene expression levels in individual microarrays. Color scheme was arbitrarily assigned. (C) Volcano plot depicting the transcriptional differences between *Hipk4* knockout and wild-type testes. RNA species that exhibited a > 2 fold change in abundance and had a P value<0.05 are shown in orange (down-regulated) or blue (up-regulated).

The online version of this article includes the following source data for figure 5:

**Source data 1.** Testis microarray data comparing WT and HIPK4 KO transcripts .

actin) to F-actin did not increase with *Hipk4* transduction (*Figure 6F*), suggesting that this kinase induces F-actin remodeling rather than filament disassembly into monomeric subunits. HIPK4 overexpression did not overtly alter the microtubule cytoskeleton in these cells (*Figure 6G–H*).

We then compared the phosphoproteomes of NIH-3T3 cells transduced with wild-type HIPK4, the K40S mutant, or a second inactive mutant that is incapable of autophosphorylation-dependent activation (Y175F) (*van der Laden et al., 2015*; *Figure 6I–J*). Through phosphopeptide enrichment, isobaric labeling, and quantitative mass spectrometry, we identified 6941 phosphosites, 303 of which increased in abundance by $\geq$2 fold upon HIPK4 overexpression (in comparison to either inactive mutant) (*Figure 6K* and *Figure 6—source data 1*). Consistent with the effects of HIPK4 overexpression on F-actin dynamics, several of these phosphosites reside in known actin-interacting proteins. For example, we identified HIPK4-dependent phosphosites in talin 1 (TLN1), AHNAK nucleoprotein, coronin 1B (CORO1B), A-kinase anchor protein 2 (AKAP2), formin 1 (FMN1), vinculin (VCL), MARCKS, paxillin (PXN), WASH family members (WASL, WIPF1, and FAM21), zyxin (ZYX), unconventional myosin 5a (MYO5a), filamins (FLNA and FLNB), and transgelins (TAGLN and TAGLN3).

## Loss of HIPK4 alters actin structures in spermatid heads

Because HIPK4 overexpression altered stress fiber dynamics in cultured cells, we hypothesized that HIPK4 may modulate F-actin-related functions in developing sperm. In particular, a role for HIPK4 in regulating acroplaxome structure and/or function could explain the head defects observed in *Hipk4*$^{-/-}$ spermatids. We first used fluorescently labeled phalloidin to visualize F-actin structures in wild-type and *Hipk4*$^{-/-}$ testis sections; however, since phalloidin predominantly labeled F-actin bundles associated with the basal and apical ectoplasmic specializations of Sertoli cells, it was difficult to discern structures within the germ cells (*Figure 7—figure supplement 1A*).

To image the acroplaxome and other anterior head structures of wild-type and *Hipk4*$^{-/-}$ spermatids, we analyzed germ cells that were enzymatically dissociated from testis tissues. Beginning at step five the acrosomes of *Hipk4*$^{-/-}$ spermatids were frequently fragmented and did not completely cover the acroplaxome (*Figure 7A*). In contrast, all wild-type spermatids had intact acrosomes that

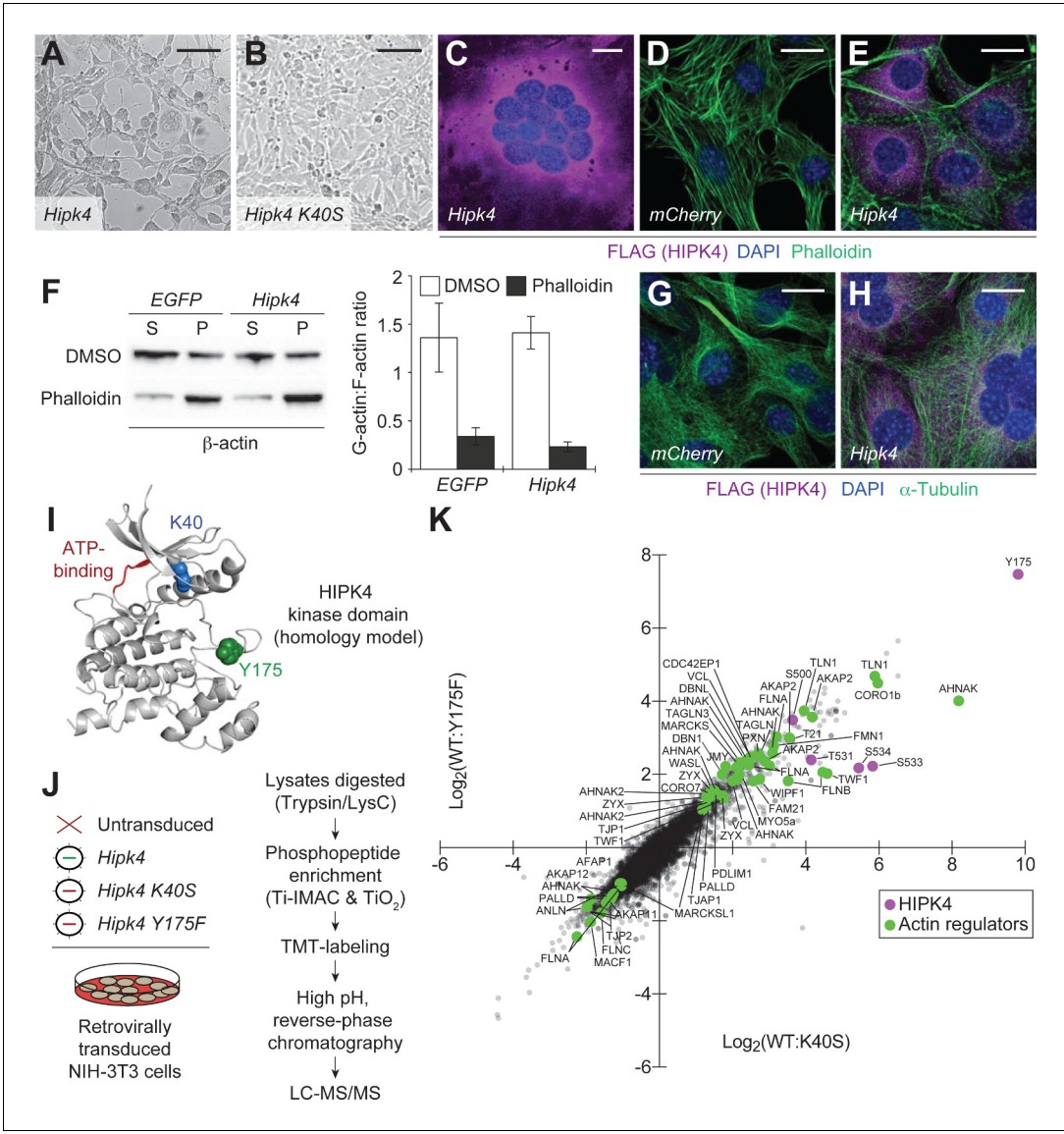

**Figure 6.** HIPK4 overexpression remodels the actin cytoskeleton in cultured cells. (**A–B**) Brightfield images of NIH-3T3 cells retrovirally transduced with FLAG-tagged *Hipk4* or kinase-dead *Hipk4 K40S*. (**C**) HIPK4-expressing NIH-3T3 cell with multiple nuclei. (**D–E**) Phalloidin and anti-FLAG staining of NIH-3T3 cells transduced with mCherry or FLAG-tagged *Hipk4*. (**F**) G- and F-actin levels in NIH-3T3 cells transduced with *EGFP* or FLAG-tagged *Hipk4* and then treated with DMSO or phalloidin after lysis. The western blot shows the soluble (S; G-actin) and pelleted (P; F-actin) pools of actin after ultracentrifugation, and the graph depicts the average G-actin:F-actin ratios of quadruplicate samples ± s.e.m. (**G–H**) α-Tubulin and FLAG immunostaining of NIH-3T3 cells transduced with *mCherry* or FLAG-tagged *Hipk4*. (**I**) Homology model of the HIPK4 kinase domain using the DYRK1A structure as a template (PDB ID: 3ANQ). The ATP-binding site is colored red, and residues that are essential for catalytic activity are depicted as blue or green space-filling models. (**J**) Work flow used to characterize the HIPK4-dependent phosphoproteome in NIH-3T3 cells. (**K**) Scatter plot of 6941 phosphosites identified by LC-MS/MS, graphed according to their relative levels in NIH-3T3 cells overexpressing wild-type or catalytically inactive HIPK4. Selected HIPK4-regulated phosphosites in actin-modulating proteins are shown in green and phosphosites in HIPK4 are shown in purple. Scale bars: A-B, 100 μm; C-E and G-H, 20 μm.

The online version of this article includes the following source data for figure 6:

**Source data 1.** Phosphoproteome of HIPK4-expressing NIH-3T3 cells.

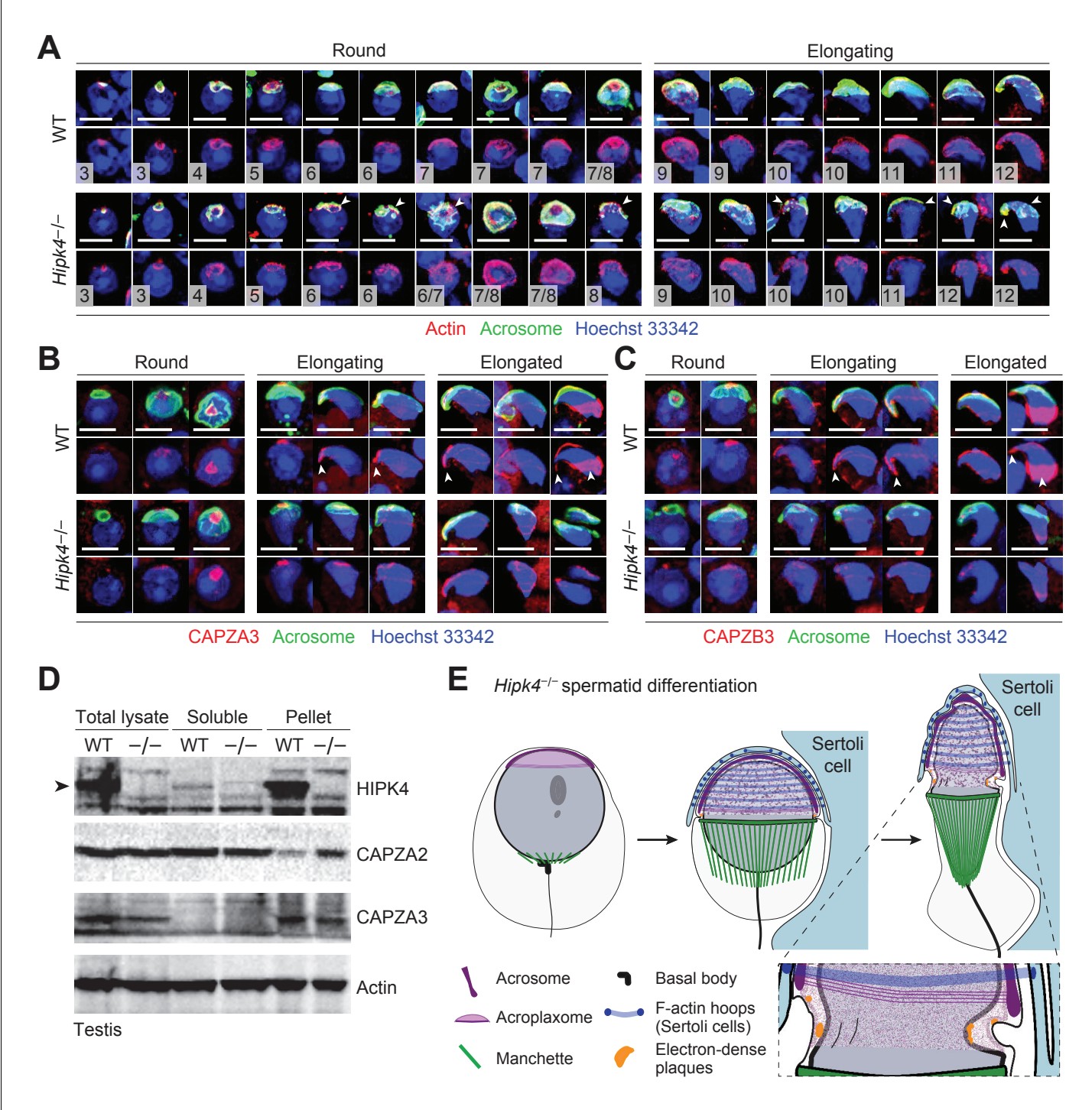

**Figure 7.** Loss of HIPK4 function alters actin dynamics in the acroplaxome. (A–C) Fluorescence imaging of enzymatically dissociated spermatids co-labeled with FITC-PNA and (A) anti-bβ actin, (B) anti-CAPZA3, or (C) anti-CAPZB3. Nuclei were stained with Hoeschst 33342. Spermatid steps are numbered in (A), and arrowheads point to fragmented acrosomes. Arrowheads in (B–C) indicate the sharp hooked apical tip of wild-type spermatids that correspond to the perforatorium and postacrosomal sheath regions. (D) Western blot of testis lysates before and after F-actin isolation by ultracentrifugation, demonstrating that HIPK4 co-precipitates with F-actin and that HIPK4 deficiency increases CAPZA2 levels in the F-actin pellet. A representative blot from three biological replicates is shown. (E) A model illustrating that HIPK4 is required to maintain the connection between the acrosome, acroplaxome, and nuclear lamina during spermatid elongation. We hypothesize that loss of HIPK4 reduces the scaffolding function of the acroplaxome, compromising its ability to withstand forces applied by the F-actin hoops of Sertoli cells. Scale bars: A-C, 2 μm.

The online version of this article includes the following figure supplement(s) for figure 7:

*Figure 7 continued on next page*

*Figure 7 continued*

**Figure supplement 1.** Phalloidin staining of F-actin dynamics in the acroplaxome.

encompassed the anterior nuclear pole. Anti-β-actin staining revealed that the fragmented acrosomes in *Hipk4*$^{-/-}$ spermatids was coincident with cytoskeletal defects in the underlying acroplaxome, and the acrosome–acroplaxome complex was extensively disassociated in some mutant cells. By treating the isolated germ cells with fluorescently labeled phalloidin, we observed that wild-type, but not *Hipk4*$^{-/-}$ spermatids, maintained F-actin at later stages of differentiation (e.g., elongated spermatids) (*Figure 7—figure supplement 1B*). Anti-β-actin and phalloidin staining also revealed severe malformation or loss of the perforatorium in elongating and elongated mutant spermatids (*Figure 7A* and *Figure 7—figure supplement 1B*), consistent with our observations of epididymal sperm by SEM. The perforatorium is a unique compartment that surrounds and extends beyond the apical tip of the nucleus in falciform spermatozoa, and it contains some the first proteins that interact with the ooplasm following fertilization (*Oko et al., 1990*; *Protopapas et al., 2019*).

To further compare the actin structures in wild-type and *Hipk4*$^{-/-}$ spermatids, we assessed the localization of actin-capping proteins CAPZA3 and CAPZB3 by immunofluorescence microscopy (*Figure 7B–C*). Wild-type and *Hipk4*$^{-/-}$ spermatids had similar expression levels and subcellular distributions of CAPZA3 and CAPZB3 during the initial stages of spermiogenesis. However, these actin regulators were selectively diminished in the perforatorium and postacrosomal sheath of elongated *Hipk4*$^{-/-}$ spermatids. In addition to these imaging studies, we investigated whether HIPK4 can co-precipitate with F-actin in testis lysates fractionated by ultracentrifugation. HIPK4 predominantly localized to the F-actin-containing pelleted fraction, suggesting that this kinase exists primarily in a complex with the actin cytoskeleton or associated proteins (*Figure 7D*). Consistent with our studies of HIPK4-overexpressing NIH-3T3 cells, HIPK4 deficiency did not appear alter G-actin:F-actin ratios in the testis lysates. However, we observed that loss of HIPK4 expression increased levels of the actin capping protein CAPZA2 in the F-actin pellet (*Figure 7D*), even though *Capza2* transcripts are downregulated in *Hipk4*$^{-/-}$ testes (*Figure 5—source data 1*). Notably, levels of F-actin-associated CAPZA3 were not HIPK4-dependent. Together, these results support a role for HIPK4 in actin remodeling during spermatid differentiation.

## Discussion

HIPK4 is a dual-specificity kinase that is expressed in male germ cells during spermatid elongation. It was previously reported that HIPK4 deficiency in mice can alter sperm morphology and number. Our studies reveal both the cell biological basis and the reproductive consequences of these spermatogenic phenotypes. Our findings establish HIPK4 as an important regulator of the acrosome–acroplaxome complex during spermiogenesis. *Hipk4*$^{-/-}$ spermatids form abnormal anterior head structures as their acrosomes become uncoupled from the underlying F-actin- and keratin 5-scaffolded acroplaxome. This is most evident by the grossly enlarged groove belt in elongating *Hipk4*$^{-/-}$ spermatids (steps 9–12). The resulting spermatozoa have misshapen heads, and a large fraction exhibit slightly irregular tail morphologies. *Hipk4*$^{-/-}$ males have lower concentrations of epididymal sperm, likely due to increased apoptosis after spermiation, and these cells have lower motility than their wild-type counterparts. Together, these spermatogenic defects closely mirror those observed in men with severe OAT syndrome, and similar to these clinical cases, HIPK4-deficient male mice are sterile.

We postulate that the abnormal head structures of HIPK4-deficient sperm are a primary cause of sterility. Sperm produced by heterozygous *Hipk4* mutant males have lower than normal epididymal concentrations of sperm and reduced total motility, but the fertility of these animals is comparable to that of wild-type mice. Only homozygous *Hipk4* mutant germ cells exhibit head defects that become increasingly overt during spermiogenesis. Consistent with diminished head functions, *Hipk4*$^{-/-}$ sperm are incompetent for IVF and have reduced binding and penetration of COCs. This diminished fertility could reflect structural defects that physically abrogate sperm-egg interactions and/or concomitant perturbations that disrupt molecular processes within the head. The hydrodynamically inefficient heads of *Hipk4*$^{-/-}$ sperm could account for their greater motility deficits, which would be expected to affect the fertility within the female reproductive tract. While it is possible

that HIPK4 contributes to the formation or function of components required for normal tail motion, thus far, our studies have not revealed flagellar defects in mutant germ cells.

Our findings indicate that the head defects of $Hipk4^{-/-}$ sperm stem from an essential role for HIPK4 in acroplaxome function during spermatid differentiation. Both optical and electron microscopy reveal defects in this matrix upon HIPK4 loss. Although the acrosome–acroplaxome complex appears to initially form correctly, expanding synchronously with the nuclear lamina during early steps of spermiogenesis (steps 4–7; *Figure 4*, *Figure 4—figure supplement 2A–C*), by step 5, round spermatids isolated from $Hipk4^{-/-}$ testes can display fragmented acrosomes that are not fully tethered to the acroplaxome. Analogous defects are evident in testis sections after spermatid elongation begins (step 8), during which Sertoli cells apply external forces to shape the spermatid head. At step 8 and beyond, the acroplaxome and nuclear lamina remain synchronous, and both structures are still juxtaposed with the perinuclear ring of the manchette. In contrast, the acrosome fails to expand posteriorly to a comparable extent at these later steps. We interpret these findings as evidence that nuclear lamina expansion proceeds normally in $Hipk4^{-/-}$ spermatids, and uncoupling of acrosome and acroplaxome dynamics is the primary defect.

Our investigations also reveal potential mechanisms by which HIPK4 could regulate the acrosome–acroplaxome complex. HIPK1-3 can directly phosphorylate homeodomain transcription factors (*Kim et al., 1998*; *Rinaldo et al., 2008*), and the structurally related dual-specificity tyrosine-regulated kinase 1a (DYRK1A) targets various transcriptional activators and repressors (*Di Vona et al., 2015*; *Litovchick et al., 2011*; *Mao et al., 2002*; *Woods et al., 2001*; *Yang et al., 2001*). Although it has been reported that HIPK4 can phosphorylate p53 in vitro (*Arai et al., 2007*; *He et al., 2010*), it is unlikely that HIPK4 acts primarily through transcriptional control. In contrast to other HIPK family members, HIPK4 lacks the homeobox-interacting domain and a nuclear localization sequence, and accordingly, it localizes to the cytoplasm rather than the nucleus (*van der Laden et al., 2015*). Moreover, our genome-wide microarray analyses reveal that loss of HIPK4 function does not lead to major transcriptional changes within the testis. These observations suggest that HIPK4 may regulate the acrosome–acroplaxome complex through more direct biochemical mechanisms.

Our cell biological studies strongly implicate HIPK4 in F-actin remodeling. HIPK4 expression in cultured somatic cells promotes the formation of branched F-actin structures and alters the phosphorylation state of multiple actin-crosslinking proteins. HIPK4 could similarly modulate F-actin networks within the acroplaxome. Consistent with this model, HIPK4 deficiency disrupts the levels of F-actin and actin-capping proteins CAPZA3 and CAPZB3 localized to the acroplaxome marginal ring and perforatorium in elongating spermatids, likely due to cytoskeletal dysregulation at earlier steps. In addition, HIPK4 co-precipitates with F-actin in testis lysates, and loss of HIPK4 expression increases the association of F-actin with CAPZA2, but not CAPZA3 in these tissue isolates. Identifying HIPK4 substrates in round spermatids will be an important next step toward understanding how this kinase regulates actin dynamics in germ cells. We also note that transcripts encoding actin-related proteins are among the few genes that are expressed in a HIPK4-dependent manner within the testis (*Figure 5—source data 1*). For example, cytoskeletal components *Actr6* and *Capza2* were downregulated in $Hipk4^{-/-}$ testis tissues, and upregulated genes included actin-membrane cross-linkers (*Tln2*, *Flnb*, and *Ank2*), actin-based motors (*Myo5b* and *Myo10*), and components of the acrosome matrix (*Zp3r* and *Zan*). We speculate that these transcriptional changes reflect a cellular response to cytoskeletal defects caused by HIPK4 deficiency.

Based on these findings and our model of HIPK4 function, we hypothesize that that HIPK4 regulates acroplaxome stability and dynamics to ensure that this cytoskeletal plate can distribute external forces applied by Sertoli cells. Acroplaxome abnormalities in $Hipk4^{-/-}$ germ cells likely emerge at the round spermatid stage and become manifest when mechanical forces are applied (e.g., Sertoli cell forces or testis dissociation) (*Figure 7E*). In particular, HIPK4 appears to promote adhesion between the acroplaxome and the overlying acrosome, as well as the formation of critical head structures such as the perforatorium and proacrosomal sheath. Loss of HIPK4 therefore leads to severe anterior malformations that preclude oocyte binding and fertilization. These head defects likely interfere with other aspects of sperm development and function, including the expulsion of excess cytoplasm, spermiation, acrosome exocytosis, and motility. In aggregate, these structural and functional deficits account for the complete sterility of HIPK4-deficient male mice. How HIPK4 regulates acrosome–acroplaxome synchrony is yet to be determined. Loss of HIPK4 did not alter the localization of several known LINC components of the acrosome and nuclear lamina. However, we note the presence

of electron dense regions within acroplaxome that appear to abut the acrosome and nuclear lamina in a HIPK4-dependent manner, raising the possibility that these structures help stabilize the acrosome–acroplaxome complex.

HIPK4 joins the small list of protein kinases that are enriched or exclusively expressed in haploid male germ cells and required for spermatogenesis. For example, the kinases CSNK2A2, SSTK, and CAMKIV function in the spermatid nucleus and promote the histone-to-protamine transition (*Escalier et al., 2003*; *Spiridonov et al., 2005*; *Wu et al., 2000*). TSSK1 and 2 are testis-specific kinases that regulate the chromatoid body, centrioles, and the developing sperm flagellum (*Jha et al., 2013*; *Kueng et al. (1997)*; *Shang et al., 2010*; *Xu et al., 2008*), and TSSK4 and TSSK5 localize to the flagellum and enable sperm motility (*Wang et al., 2016*; *Wang et al., 2015*). Few kinases have been implicated in spermatid head shaping, and interestingly, several are truncated forms of tyrosine protein kinases. For example, TR-KIT, a splice variant of c-KIT is expressed during spermiogenesis and thought to phosphorylate PLCγ1. Ectopic expression of a constitutively active TR-KIT mutant causes manchette dysregulation. TR-KIT is also present in sperm, and serves to activate oocytes during fertilization by promoting the completion of meiosis II and formation of the pronucleus. A truncated, testis-specific version of FER, FERT, is localized to the acroplaxome and phosphorylates cortactin, regulating actin polymerization/degradation (*Keshet et al., 1990*; *Kierszenbaum et al., 2008*). However, male mice lacking FERT are still fertile (*Craig et al., 2001*), perhaps due to redundancy between kinase substrates. To the best of our knowledge, HIPK4 is the first kinase known to be essential for acrosome–acroplaxome function and male fertility.

Given the druggable nature of kinases, HIPK4 is also a new potential target for male contraception. Small-molecule HIPK4-specific antagonists would be predicted to recapitulate the male sterility caused by disrupting the *Hipk4* gene in mice. Such treatments would reduce male fertility without interfering with the hypothalamus-pituitary-gonadal axis, damaging somatic cells of the testis, or introducing genetic abnormalities to offspring if fertilization is achieved. Since HIPK4 expression is restricted to later steps of sperm development, HIPK4 inhibitors also would induce sterility more quickly than drugs that perturb earlier steps in spermatogenesis. Contraceptive reversibility would be equally rapid. Moreover, HIPK4 antagonists could have minimal non-reproductive effects, as $Hipk4^{-/-}$ mice appear to have otherwise normal physiology. Further investigations of HIPK4 therefore could not only elucidate the mechanisms that drive spermatid differentiation but also address a longstanding unmet need in reproductive medicine.

## Materials and methods

Key Resource Table (see *Supplementary file 1* – Key Resources Table).

### Ethics statement

All animal studies were conducted in compliance with the Stanford University Institutional Animal Care and Use Committee under Protocol 29999. Vertebrate research at the Stanford University School of Medicine is supervised by the Department of Comparative Medicine's Veterinary Service Center. Stanford University animal facilities meet federal, state, and local guidelines for laboratory animal care and are accredited by the Association for the Assessment and Accreditation of Laboratory Animal Care International. De-identified human testis sections were obtained from Stanford University in compliance with protocol IRB-32801.

### Animal use

$Hipk4^{+/tm1b}$ breeding pairs (C57BL/6NJ background) and wild-type (C57BL/6NJ) mice were purchased from The Jackson Laboratory. Mice used for this study were weaned at 19–22 dpp and genotyped using Platinum *Taq* DNA Polymerase (Invitrogen) and the following primers: wild-type forward 5'-CCTTTGGCCTTATACATGCAC-3', wild-type reverse 5'-CAGGTGTCAGGTCTGGCTCT-3', mutant forward 5'-CGGTCGCTACCATTACCAGT-3', mutant reverse 5'-ACCTTGAGATGACCCTCCTG-3'.

### Cell line use

The cell line HEK-293T (*Homo sapiens*), ATCC, cat no. CRL-3216, cells were used by passage 7 in all experiments for this paper, mycoplasma-free (Lonza, LT07-118, control: LT07-518). The cell line NIH-

3T3 (Mus muscles), ATCC, cat no. CRL-1658, used by passage 5 in all experiments for this paper, mycoplasma-free (Lonza, LT07-118, control: LT07-518)

## Tissue distribution of Hipk4

TaqMan primers (Applied Biosystems) for *Hipk4* (Mm01156517_g1) were used to probe the Origene TissueScan Mouse Normal cDNA Array according to the manufacturer's protocols. Gene expression levels were normalized to *GAPDH* (Mm99999915_g1).

## In situ hybridization analysis

Whole testes were dissected and immediately fixed in freshly prepared modified Davidson's fixative (30% formaldehyde,15% ethanol, 5% glacial acetic acid, 50% distilled water) for 16–20 hr and then washed and stored in 70% ethanol until further use. For in situ hybridization analyses, the fixed tissues were paraffin-embedded, cut into 10 μm sections, and mounted on slides. To detect *Hipk4* transcripts, we used the RNAscope 2.5 HD Detection Kit (Advanced Cell Diagnostics) with *Hipk4* probes (Mm-Hipk4, 428071), following the manufacturer's protocol for formalin-fixed, paraffin-embedded (FFPE) sections. Hybridization probes for *Ppib* (BA-Mm-Ppib-1ZZ, 313911) and *dapB* (BA-DapB-1ZZ, 310043) were used as positive and negative controls, respectively.

## Assessment of fecundity

To test fertility by mating, we paired $\geq$7 week-old males (wild-type, *Hipk4*$^{+/-}$, and *Hipk4*$^{-/-}$) with age-matched, wild-type females for 18–21 days, and the number of live-born pups for each pairing was recorded. Male and female mice were paired in this manner for 2–4 rounds (6–12 weeks).

IVF and ICSI procedures were performed at the Transgenic, Knockout, Tumor Model Center at Stanford University. For IVF studies, C57BL/6NJ females were superovulated by injection with PMSG (ProSpec, 5U, 61–63 hr prior to oocyte-harvesting) and hCG (ProSpec, 5U, 48 hr after PMSG injection). On the day of the experiment, epididymal sperm were isolated using a 'swim-out' method in TYH medium (120 mM NaCl, 5 mM KCl, 2.5 mM $MgSO_4$, 1.0 mM $KH_2PO_4$, 25 mM $NaHCO_3$, 2.5 mM $CaCl_2$, 1 mM sodium pyruvate, 1.0 mg/mL glucose, 1.0 mg/mL methyl-β-cyclodextrin, and 1.0 mg/mL polyvinylalcohol; pH 7.2) and transferred to TYH medium containing bovine serum albumin (BSA; final concentration of 5 μg/mL). Cumulus-oocyte complexes (COCs) were then harvested from the oviducts of these superovulated females in M2 medium with HEPES buffer (Sigma, M7167), and transferred through three 50 μL drops of HTF medium (Millipore, MR-070-D) containing 0.25 mM reduced glutathione under mineral oil, pre-equilibrated to 37°C, 5% $CO_2$. After 1 hr of capacitation, motile sperm ($\sim$5.0$\times$10$^5$) were added to the drop of HTF medium containing COCs and incubated for 4 hr at 37°C and 5% $CO_2$. Sperm-oocyte complexes were then washed four times in M2 medium and incubated in 30 μL KSOM medium (Millipore, MR-101-D) at 37°C, 5% $CO_2$ under mineral oil. The number of two-cell, morula, and blastocyst-stage embryos were then counted over the next 72 hr.

ICSI experiments were performed as previously described (*Yoshida and Perry, 2007*). Briefly, motile sperm were harvested from the epididymis, and sperm heads were injected into the cytoplasm of CD1 oocytes using a piezo-actuated micromanipulator. The injected embryos were cultured in KSOM medium at 37°C. After 24 hr, live two-cell embryos were either cultured until the blastocyst stage or implanted into the oviducts of pseudo-pregnant female mice.

## Antibodies

The following primary antibodies were used for western blotting and immunofluorescence imaging: anti-HIPK4 (FabGennix International, rabbit pAb generated against the peptide sequence PAGSKSD SNFSNLIRLSQVSPED); anti-KPNB1 (H-300) (Santa Cruz Biotechnology, rabbit pAb); anti-phosphotyrosine (Upstate/Millipore, 4G10 Platinum, rabbit pAb); anti-ZP3R/mouse sp56 (7C5) (QED Bioscience,, mouse mAb); anti-IZUMO1 (125) (Abcam, rat mAb); anti-SPACA1 (Abcam, rabbit pAb); anti-FLAG (M2) (Sigma, mouse mAb); anti-α-tubulin (3H3085) (Santa Cruz Biotechnology, rat mAb); anti-β-actin (Cytoskeleton, rabbit pAb), anti-CAPZA3 (Progen, guinea pig pAb); anti-CAPZB3 (Progen, guinea pig pAb); anti-CAPZA2 (ProteinTech, rabbit pAb); anti-DPY19L2 (gift from Christophe Arnoult, rabbit pAb), anti-SUN1 (gift from Manfred Alsheimer, guinea pig pAb); anti-nesprin3 (gift from Arnoud Sonnenberg, rabbit pAb); Alexa Fluo$^r$ 488-conjugated anti-EB3 (EPR11421-B) (Abcam, rabbit pAb).

Secondary antibodies included HRP-conjugated sheep anti-mouse IgG (GE Healthcare), HRP-conjugated donkey anti-rabbit IgG (GE Healthcare), Alexa Fluor 594-conjugated anti-guinea pig (Invitrogen), Alexa Fluor Plus 647-conjugated goat anti-mouse IgG (Invitrogen), Alexa Fluor 647-conjugated goat anti-rat IgG (Invitrogen), and Alexa Fluor Plus 647-conjugated goat anti-rabbit IgG (Invitrogen).

## Western blot analyses

To detect HIPK4 protein, testis lysates were prepared by sonication of ~100 mg pieces of freshly removed whole testis in 1.0 mL ice-cold RIPA buffer containing protease inhibitors (cOmplete, EDTA-free Protease Inhibitor Cocktail Tablets, Roche) and phosphatase inhibitors (PhosSTOP, Roche). Equivalent amounts (12 µg) of total protein per sample were diluted with 6x Laemmli sample buffer, boiled for 5 min, and stored at −20°C until use. They were then loaded on a 3–8% tris-acetate gel (Bio-Rad) for SDS-PAGE. Proteins were transferred to a PVDF membrane and immunoblotted at 4°C overnight with the following primary antibodies: anti-HIPK4 [0.55 µg/mL, 1:1000 dilution in phosphate buffered saline (PBS) containing 0.1% Tween 20% and 4% BSA] or anti-KPNB1 (1:1,000). Chemiluminescence detection was conducted with HRP-conjugated secondary antibodies (1:20,000) and either SuperSignal West Dura or SuperSignal Femto kits (Pierce/Thermo Fisher Scientific).

## Assessment of sperm quality

Epididymides from male mice (8–9 weeks old) were dissected and cleared of fat tissue before being cut open with fine scissors in 200 µL HTF medium equilibrated at 37°C and 5% $CO_2$. The sperm suspension (10 µL) was counted and assessed for total motility at 100x magnification using Leja slides and a microscope equipped with a reticle. Sperm quality parameters of males (15–17 weeks old) were also assessed by the Mouse Biology Program at UC-Davis using the following procedures. Morphology was assessed visually, and CASA was used to confirm sperm concentrations and motility. Sperm were viewed randomly under 400x or 1000x magnification to determine the percentage with abnormal head sizes and shapes (macrocephalous, microcephalous, tapered, triangular, olive, pin, banana, amorphous, collapsed, abnormal hook, irregularly shaped, etc.) or abnormal midpieces or tails (bent, coiled, short, thin, crinkles, irregularly shaped, etc.). Sperm were not included in the morphology assessment if they were: (1) aggregated; (2) had a back or abdomen view of the head; or (3) were decapitated. At least 100 sperm in 5 or more fields of view (at least 500 in total) were evaluated for each experimental condition.

## Assessment of sperm maturation and function

To assess capacitation by western blot, motile sperm were collected in 2.5 mL of TYH medium (BSA-free, pH 7.4, and $CO_2$-equilibrated) from cauda epididymides using the 'swim out' method at 37°C and 5% $CO_2$. The sperm suspension was then centrifuged at 600 $g$, and all but 0.4 mL of the supernatant was removed to achieve a final concentration of ~$40 \times 10^6$ cells/mL. In fresh polystyrene tubes, 200 µL of the sperm solution was added to 1.0 mL TYH medium containing BSA (10 mg/mL), and the suspension was incubated at 37°C for 1.5 hr. The sperm were then centrifuged at 13,000 $g$ for 1 min, washed with PBS containing phosphatase inhibitors, and pelleted once more. After the addition of 1x Laemmli sample buffer without reducing agent (25 µL), and the samples were boiled for 5 min and centrifuged at 13,000 $g$. The supernatants were collected and 2 µL of β-mercaptoethanol was added to each sample, which were then boiled again for 1 min. Following SDS-PAGE, proteins were transferred to a PVDF membrane and immunoblotted with an anti-phosphotyrosine antibody (1:1000 dilution, Upstate/Millipore) for chemiluminescence detection.

To perform in vitro acrosome reactions, sperm were collected and capacitated as described above. Following 1.5 hr of capacitation, the $Ca^{2+}$ ionophore A23187 (hemicalcium salt; Sigma; 1000x stock dissolved in ethanol) was added to the suspensions to achieve final concentrations of 10 µM. After incubation for 1 hr at 37°C and 5% $CO_2$, and the cells were pelleted by centrifugation at 600 $g$, and all but 0.5 mL of the supernatant was removed. The following steps were then performed in microcentrifuge tubes, and the cells were pelleted at 600 $g$ and washed with PBS between each step. Sperm were fixed by adding 2.0 mL 4% paraformaldehyde in PBS for 15 min and permeabilizing with 0.3% Triton-X100 in PBS for 5 min. For experiments measuring acrosome exocytosis, the sperm were then incubated for 30 min in PBS containing 2% BSA, 0.01% Triton X-100, and fluorescein-conjugated peanut agglutinin (FITC-PNA, 10 µg/mL, Sigma) and 10 µg/mL Hoechst 33342

nuclear stain. After washing the cells with PBS, they were mounted on microslides with Vectashield Vibrance medium (Vector Labs). For immunolabeling experiments, the cells were then divided into separate tubes, blocked for 1 hr with PBS containing 2% BSA and 0.01% Triton X-100, incubated for 1 hr at room temperature with primary antibodies (10 µg/mL final concentration in blocking buffer), and then washed once with blocking buffer. The samples were then incubated with fluorescently labeled secondary antibodies and 10 µg/mL FITC-PNA in blocking buffer for 30 min. The sperm were washed once more, transferred to microscope slides, mounted with Prolong Gold medium with DAPI, and then imaged on a Zeiss LSM 700 confocal microscope equipped with a 63x oil-immersion objective. ImageJ software (NIH) was used to create maximum-intensity Z-stack projections, and Photoshop CS6 (Adobe) was used to crop images and adjust fluorescence intensity levels.

To assay for oocyte binding in vitro, we followed the same general protocol described for IVF experiments. After 3 hr of sperm-COC incubation and five washing steps, we transferred the complexes in a small volume into a 50 µL drop of PBS containing 4% paraformaldehyde on a microscope slide. After 30 min, excess liquid was carefully removed with a Kimwipe, and the slides were mounted with Prolong Gold medium with DAPI. Fluorescence and DIC imaging on a Leica DMi8 microscope at 200x magnification were used to assess the number of sperm bound to oocytes.

## TUNEL assays

To measure DNA double strand breaks by terminal deoxynucleotidyl transferase-mediated dUTP nick end labeling (TUNEL), we used the In Situ Cell Death Detection Kit (Sigma-Aldrich), following the manufacturer's protocol (ver. 17) for cell suspensions. Fluorescence imaging was performed using a Zeiss LSM 700 confocal microscope equipped with a 63x oil-immersion objective. ImageJ software (NIH) was used to create maximum-intensity Z-stack projections and TUNEL-positive cells were manually counted (n $\geq$ 357).

## Histology of testis and epididymis

Histological staining (PAS and H and E) was performed on formalin-fixed, paraffin-embedded testis and epididymis sections (10 µm). To obtain these sections, the tissues were removed from 12- to 15-week-old males, immediately fixed in modified Davidson's fixative for 16 hr, and stored in 70% ethanol until paraffin-embedding, sectioning, and mounting on microscope slides. The slides then were heated for 1 hr at 50°C, washed with xylenes, re-hydrated, and stained with periodic acid solution (Sigma) and Schiff's reagent (Sigma) and/or counterstained with modified Harris hematoxylin solution (Sigma) and Eosin Y (Sigma).

## Electron microscopy

For scanning electron microscopy, epididymal sperm were collected in TYH medium by the 'swim-out' method at 37°C, pelleted at 200 $g$, resuspended in 0.1 M sodium cacodylate buffer (pH 7.4) containing 2% paraformaldehyde and 4% glutaraldehyde, and transferred to individual wells of a 24-well plate containing poly-D-lysine-coated 12-mm coverslips. The samples were allowed to fix overnight at 4°C and then post-stained with 1% aqueous osmium tetroxide (EMS, 19100) for 1 hr. $OsO_4$-treated samples were rinsed in ultrafiltered water three times and gradually dehydrated in increasing concentrations of ethanol (50%, 70%, 90%, 2 × 100%; 15 min each). Each coverslip was then dried at the critical point with liquid $CO_2$ using a Tousimis Autosamdri−815A system and 15-min purge time. Dried samples were sputter-coated (100 Å, Au/Pd) before imaging with a Zeiss Sigma FE-SEM using In-Lens and lateral Secondary Electron detection at 3.02 kV.

For transmission electron microscopy, samples were fixed in 0.1 M sodium cacodylate buffer (pH 7.4) containing 2% glutaraldehyde and 4% paraformaldehyde at room temperature for 1 hr. The fixative was then replaced with cold aqueous 1% $OsO_4$, and the samples were allowed to warm to room temperature for 2 hr, washed three times with ultrafiltered water, and stained in 1% uranyl acetate for 2 hr. Samples were then dehydrated in a series of ethanol washes, beginning at 50%, then 70% ethanol, and then moved to 4°C overnight. They were then placed in cold 95% ethanol and allowed to warm to room temperature, changed to 100% ethanol for 15 min, and finally to propylene oxide (PO) for 15 min. Samples were next incubated with sequential EMbed-812 resin (EMS, 14120):PO mixtures of 1:2, 1:1, and 2:1 for 2 hr each and stored overnight in 2:1 resin:PO. The samples were then placed into 100% EMbed-812 resin for 4 hr, moved into molds with fresh resin, orientated and

warmed to 65°C overnight. Sections were taken between 75 and 90 nm, picked up on formvar/carbon-coated slot Cu grids, stained for 40 s in 3.5% uranyl acetate in 50% acetone, followed by staining in 0.2% lead citrate for six minutes. The sections were then imaged using a JEOL JEM-1400 120 kV instrument and a Gatan Orius 832 4k × 2.6 k digital camera with 9 μm pixel size.

## Microarray assay

Three testes from 12-week-old wild-type and *Hipk4*$^{-/-}$ mice were snap frozen in liquid N$_2$, and stored at −80°C. Samples were thawed and 50 mg portions were immediately homogenized in TRIzol, and RNA was isolated using the Direct-zol RNA MiniPrep Plus kit (Zymo Research, R2070S) and stored at −80°C. RNA profiling was then conducted in quadruplicate Mouse Clariom D assays (Applied Biosystems, 902514), following the manufacturer's protocol. Briefly, 50 ng of each sample was used as an input into the GeneChip WT Plus Reagent Kit, and labeled targets were hybridized to arrays in a GeneChip Hybridization Oven 645. Washing and staining steps were performed on a GeneChip Fluidics Station 450, and arrays scanned on a GeneChip Scanner 3000 7G system. Data were analyzed using the Transcriptome Analysis Console 4.0 software package.

## Retrovirus production

Murine *Hipk4* and the *K40S* mutant genes were obtained by PCR using the primers 5′-CAAAAAAG-CAGGCTCAGCCACCATGGCCACCATCCAGTCAGAGACTG-3′ and 5′-CAAGAAA GCTGGGTCG TGGTGCCCTCCAACATGCTGCAG-3′ for wild-type *Hipk4* and 5′-TCGATCCTGA AGAACGATGCG TACCGAAGC-3′ and 5′-GATGGCCACCATTTCACCTGTACTCCGAC-3′ for the *Hipk4-K40S* mutant. pCL-ECO was purchased from Imgenex, and pBMN-I-GFP was provided by Gary Nolan. For Gateway recombination-mediated cloning, the PCR products were amplified further with the primers 5′-GGGGACAAGTTTGTACAAAAAAGCAGGCTCA-3′ and 5′-GGGGACCACTTTGTACAAGAAAGC TGGGTC-3′ and Phusion polymerase (New England Biolabs) to add attB adapter sequences. The clones were then transferred into pDONR223 in a BP recombination reaction using BP clonase II (Invitrogen) according to the manufacturer's protocols. pDONR223 entry constructs were next transferred to pBMN-3xFLAG-IRES-mCherry-DEST vectors using LR clonase II (Invitrogen) according to the manufacturer's protocols. pBMN-HIPK4-Y175F-3xFLAG-IRES-mCherry was later generated by site-directed mutagenesis of the wild type construct using the primers 5′-CGCTATGTGAAGGAGCC TTTCATCCAGTCCCGCTTCTAC-3′ and 5′-GTAGAA GCGGGACTGGATGAAAGGCTCCTTCACA TAGCG-3′ and *Pfu*Ultra II Fusion polymerase (Agilent).

Retroviral stocks were prepared from HEK-293T cells seeded in 10 cm tissue culture dishes (~4×10$^6$ cells/dish) in 10 mL of culture medium. Approximately, 18 hr post-seeding, each 10 cm dish was transfected as follows: pBMN-3xFLAG-IRES-mCherry-DEST plasmids containing wild type or mutant *Hipk4* (7.4 μg) and the pCL-ECO retrovirus packing vector (4.4 μg) were diluted in OMEM medium (375 μL). This DNA mixture was added to 40 μL Fugene HD reagent (Promega) in OMEM (335 μL) and incubated at room temperature for 15 min, before being gently added to culture medium on cells. After 24 hr, the medium was replaced with DMEM containing 10 mM HEPES (pH 7.4), 3% fetal bovine serum, 7% calf serum, and 1% sodium pyruvate. Retrovirus-containing supernatant was then collected three times at 24 hr intervals, passed through a 0.45 μm filter, and stored at −80°C.

## F-actin sedimentation assays

NIH-3T3 cells were seeded into 6-well plates at a density of 150,000 cells/well and infected 18 hr later with medium containing *Hipk4*-encoding retrovirus and polybrene (8 μg/mL), achieving a multiplicity of infection (MOI) of ~4. Lysates and sedimentation fractions were prepared 48 hr after transduction, using the reagents and protocols outlined in the G-Actin/F-actin In Vivo Assay Biochem Kit (Cytoskeleton). Briefly, the cells were lysed by sonication in LAS 1 buffer (37°C), and 100 μL aliquots were spun at 100,000 *g* for 1 hr at 37°C. The supernatant was carefully removed, and the pellets were resuspended in 100 μL LAS2 de-polymerizing buffer for 1 hr on ice. Samples were diluted with 6 SDS sample buffer, boiled for 7 min, and stored at −20°C.

To perform analogous experiments using testis lysates, testes from 12-week-old mice were dissected and decapsulated in warm DMEM containing 10 mM HEPES, pH 7.4. Collagenase Type I (final concentration of 1.5 mg/mL, Worthington Biochemical) and DNase I (1.0 mg/mL) were added,

and the seminiferous tubules were gently pulled apart with forceps and incubated at 37°C for 15 min. The supernatant was replaced with 2.0 mL of LAS1 buffer, and the resuspended tubules were lysed by sonication. F-actin and G-actin fractions were prepared by centrifugation according the manufacturer's protocols.

The NIH-3T3 and testis samples were analyzed by western blots, using the same procedures described above for detecting HIPK4 and following primary antibody conditions: anti-HIPK4 (1:1,000, incubated overnight at 4°C), anti-CAPZA2 (1:1,000, incubated overnight at 4°C), anti-CAPZA3 (1:1,000, incubated overnight at 4°C), and anti-β actin (1:1,000, incubated 1 hr at room temperature).

## Quantitative phosphoproteomics by mass spectrometry

To obtain peptides suitable for phospho-enrichment and mass spectrometry studies, NIH-3T3 cells (passage 6) were seeded in 10 cm dishes (five per condition) at $1.5 \times 10^6$ cells/dish with DMEM containing 10% calf serum, 0.1% sodium pyruvate, 100 U/mL penicillin, and 0.1 mg/mL streptomycin and transduced after 12 hr with the appropriate retrovirus and polybrene (8 µg/mL) to achieve an MOI > 4. After 48 hr, the culture medium was replaced with DMEM containing 0.1% CS for 4 hr, and then all dishes were transferred to a cold room, rinsed twice with ice-cold PBS, and incubated on a rocker for 10 min with 0.4 mL RIPA lysis buffer containing protease inhibitors (cOmplete tablets, Roche) and phosphatase inhibitors (PhosSTOP tablets, Roche; 0.2 mM PMSF, Sigma). Cells were manually scraped off of each dish, and the suspensions were transferred to 15 mL Falcon tubes on ice. Cells were sonicated for 30 s on ice, and the lysates were cleared by centrifugation at 14,000 $g$. A portion of each lysate was removed for protein concentration determination and western blot analyses. The remaining protein lysates were precipitated with 14 mL of cold acetone at −80°C. Precipitates were pelleted, the supernatant was thoroughly removed, and proteins were resolubilized with 2.0 mL of 8.0 M urea, 50 mM sodium bicarbonate (pH 8.0). Samples were reduced at room temperature for 30 min with the addition of DTT to a final concentration of 5 mM, and then alkylated in the dark using acrylamide at a final concentration of 10 mM for 30 min. To digest proteins, samples were diluted to 1 M urea with 50 mM sodium bicarbonate (pH 8.0), Protease Max Surfactant (Promega, V2072) was added to a final concentration of 0.03%, and a Trypsin–LysC protease mix (Promega, V5073) was added at ~1:40 ratio to the protein concentration of each sample. Samples were incubated at 37°C for 14 hr.

The digests were acidified to pH 3.5 with formic acid and incubated for 15 min at 37°C to break down surfactants. Peptides were purified using Oasis HLB columns (3 mL) (Waters Co., WAT094226) according to the manufacturer's protocol, lyophilized overnight, and resolubilized in a 4:1 acetonitrile/$H_2O$ mixture containing 0.1% trifluoracetic acid. Peptide concentrations were determined using the Pierce Peptide Quantification Colorimetric Assay. Samples were adjusted to 1.0 mg/mL by adding a 4:1 acetonitrile/$H_2O$ mixture containing 0.2% trifluoracetic acid, and 20 µL of the resulting solution was saved for 'total peptide' mass spectrometry runs. A standard set of phosphopeptides (4 pmol/peptide sample, MS PhosphoMix1 Light, Sigma, MSPL1) were spiked into each sample. Phosphopeptide-enrichment steps were performed using a 1:1 mix of MagReSyn Ti-IMAC and MagReSy $TiO_2$ resins (ReSyn, MR-TIM005 and MR-TID005) according to the manufacturer's protocols. The eluted peptides were acidified to pH 2.5 with 10% TFA in $H_2O$ and reduced to a volume of 5–10 µL using a SpeedVac concentrator. The samples were further enriched for hydrophilic peptides by purification through graphite spin columns (Pierce, 88302) according to the manufacturer's protocol. Samples were adjusted to pH ~8.0 with a 100 mM triethylammonium bicarbonate solution and dried on a SpeedVac concentrator. The peptides from different conditions were then isobarically labeled using the TMT-6plex kit (Pierce, 90061) according to the manufacturer's protocols and pooled together. Finally, these pooled peptide samples were fractionated into six fractions using a high-pH reversed-phase peptide fractionation kit (Pierce, 84868, lot RF231823B ), and each fraction was run on a Thermo Orbitrap Fusion Tribrid for LC-MS/MS analysis.

Peptides were identified using SEQUEST software, and individual species were removed from the analysis if they: (1) had a false discovery rate was above 1%; (2) were contaminating peptides (i.e. bovine or human); (3) were not phosphorylated. The signal intensities were then normalized for each TMT channel, based on the sum of the signals detected for the standard phosphopeptide mix that had been spiked into each sample prior to the phospho-enrichment step. Peptides that were identified as having the same phosphorylation state were combined, converging on 6947 phosphosites

with their associated A scores. The fraction of individual TMT signals relative to the total intensity for each peptide was then determined.

## Immunofluorescence studies

To determine the effects of *Hipk4* overexpression in NIH-3T3 cells, the fibroblasts were seeded into 6-well plates at a density of 150,000 cells/well and infected 18 hr later with the appropriate retrovirus at an MOI of ~4. One day after infection, cells were reseeded into 24-well plates containing poly-D-lysine-coated 12-mm glass coverslips and cultured for 24 hr in DMEM containing 10% calf serum, 100 U/mL penicillin, and 0.1 mg/mL streptomycin. The cells were then treated with DMEM containing 0.5% calf serum and the antibiotics for an additional 24 hr, fixed in PBS containing 4% paraformaldehyde for 10 min at room temperature, and washed 3 × 5 min with PBS. The cells were permeabilized with PBS containing 0.3% Triton X-100 for 5 min and blocked for 2 hr at 4°C in PBS containing 2% BSA and 0.1% Triton X-100. The cells were then incubated for 1 hr at room temperature with primary antibodies (1:200 dilution in blocking buffer), washed 3 × 5 min with PBS, incubated for 30 min with the appropriate secondary antibodies (1:400 dilution in blocking buffer) and/or Alexa Fluor 647-conjugated phalloidin (1:400, Invitrogen), and washed twice more with PBS. Nuclei were stained with DAPI, and the coverslips were rinsed briefly in water and mounted onto slides using Prolong Gold Antifade reagent (Invitrogen).

Immunofluorescence imaging of testis sections was conducted using cryosections of fresh frozen tissue (10 µm thick). Once sectioned, samples were fixed on the slides with PBS containing 4% paraformaldehyde for 30 min at room temperature, permeabilized with PBS containing 1% Triton-X-100 for 15 min, blocked with PBS containing 2% BSA and 0.01% Triton X-100, incubated with primary antibody (10 µg/mL in blocking buffer) for 2 hr, washed three times with PBS containing 0.01% Triton X-100, incubated with secondary antibody (1:400 dilution in blocking buffer) along with either Alexa Fluor 647-conjugated phalloidin (1:400) or fluorescently labeled anti-EB3 (1:100), for 30–45 min, washed three times with PBS containing 0.01% Triton X-100, and mounted with coverslips with Prolong Gold with DAPI.

Immunofluorescence imaging of isolated spermatids was conducted on cells enzymatically dissociated from seminiferous tubules. Briefly, testes were dissected and decapsulated in warm DMEM containing HEPES (10 mM, pH 7.4). Collagenase Type I (final concentration of 1.5 mg/mL, Worthington Biochemical) and DNase I (1.0 mg/mL) were added, and tubules were gently separated and incubated at 37°C for 15 min. The tubules were then transferred to DMEM, cut into small fragments using fine scissors, and incubated with trypsin (2.0 mg/mL, Worthington Biochemical) and DNase I (2.0 mg/mL) in DMEM for 20 min at 37°C with vigorous physical mixing every 4 min using a plastic transfer pipette. BSA was added to stop enzymatic digestion, and the cells were pelleted at 400 *g* for 10 min at 4°C. The cells were thoroughly resuspended in PBS containing 1 mg/mL polyvinyl alcohol (PVA) and carefully loaded onto a gradient column of BSA in DMEM containing HEPES buffer (10 mM, pH 7.4). The columns contained 25 mL zones of 4%, 3%, 2%, and 1% BSA. After 4 hr of gravity sedimentation, 12 mL fractions of cells were collected, and those containing round, elongating, and elongated spermatids were combined and pelleted at 400 *g*. Cells were suspended in the PBS/PVA buffer, pelleted, suspended in a hypotonic sucrose solution (20 mM HEPES, 50 mM sucrose, 17 mM sodium citrate) for 10 min, pelleted, and then fixed in PBS containing 4% paraformaldehyde at room temperature for 15 min. The cells were then permeabilized with PBS containing 1 mg/mL PVA and 1.0% Triton X-100, blocked with PBS containing 2% BSA and 1 mg/mL PVA, and incubated with the appropriate primary antibody (1:50 dilution in blocking buffer) at 4°C overnight with end-over-end rotation. After three 10 min washes with PBS containing 0.01% Triton X-100 and 1 mg/mL PVA at room temperature, the cells were incubated with the appropriate fluorescently labeled secondary antibody (1:400 dilution in blocking buffer) along with FITC-PNA (1:1,000), Alexa Fluor 647-conjugated phalloidin (1:400), and/or fluorescently labeled anti-EB3 (1:200) for colocalization) for 30–45 min, washed once, incubated with PBS/PVA buffer containing Hoechst 33342 dye for 10 min at room temperature, washed once more, and mounted on microscope slides using Vectashield Vibrance medium.

All fluorescence imaging was performed using a Zeiss LSM 700 or 800 confocal microscope equipped with a 63x oil-immersion objective. ImageJ software (NIH) was used to create maximum-intensity Z-stack projections, and Photoshop CS6 (Adobe) was used to crop images and adjust fluorescence intensity levels.

## Acknowledgements

This work was supported by R21 HD78385 (JKC), a Male Contraceptive Initiative Research Grant (JKC), and postdoctoral fellowships from the American Cancer Society (JAC) and the Male Contraceptive Initiative (JAC). We gratefully acknowledge Lydia-Marie Joubert (Cell Sciences Imaging Facility, Stanford University) for providing training related to our scanning electron microscopy experiments. We are also indebted to Pablo Visconti (University of Massachusetts), Moira O'Bryan (Monash University), George Gerton (University of Pennsylvania), and Michael Eisenberg (Stanford University) for their thoughtful discussions and protocols. Antibody reagents were kindly provided by Christophe Arnoult (Université Grenoble Alpes), Manfred Alsheimer (University of Würzburg), and Arnoud Sonnenberg (Universiteit Leiden). John Higgins (Stanford University) supplied human testis sections. The electron microscopy studies were supported, in part, by ARRA Award Number 1S10RR026780-01 from the National Center for Research Resources (NCRR).

## Additional information

### Competing interests

J Aaron Crapster: JAC has served as a consultant for Vibliome Therapeutics, which is developing small-molecule inhibitors of HIPK4 and other kinases, and he is now a Principal Scientist at the company. Paul G Rack: PGR is an employee of Thermo Fisher Scientific. James K Chen: JKC serves on the Scientific Advisory Board for Vibliome Therapeutics. The other authors declare that no competing interests exist.

### Funding

| Funder | Grant reference number | Author |
| --- | --- | --- |
| American Cancer Society | 125153-PF-13-377-01-DMC | J Aaron Crapster |
| Male Contraception Initiative | | J. Aaron Crapster James K Chen |
| National Institutes of Health | R21 HD78385 | James K Chen |

The funders had no role in study design, data collection and interpretation, or the decision to submit the work for publication.

### Author contributions

J Aaron Crapster, Conceptualization, Data curation, Formal analysis, Supervision, Funding acquisition, Investigation, Visualization, Methodology, Project administration; Paul G Rack, Data curation, Formal analysis, Investigation; Zane J Hellmann, Data curation, Investigation; Austen D Le, Data curation; Christopher M Adams, Conceptualization, Data curation, Investigation; Ryan D Leib, Conceptualization, Data curation, Formal analysis; Joshua E Elias, Data curation, Formal analysis; John Perrino, Resources, Investigation; Barry Behr, Conceptualization, Investigation; Yanfeng Li, Jennifer Lin, Investigation; Hong Zeng, Conceptualization, Resources, Formal analysis, Supervision; James K Chen, Conceptualization, Resources, Data curation, Formal analysis, Supervision, Funding acquisition, Visualization, Project administration

### Author ORCIDs

J Aaron Crapster (ID) https://orcid.org/0000-0001-8185-8401

### Ethics

Animal experimentation: All animal studies were conducted in compliance with the Stanford University Institutional Animal Care and Use Committee under Protocol 29999. Vertebrate research at the Stanford University School of Medicine is supervised by the Department of Comparative Medicine's Veterinary Service Center. Stanford University animal facilities meet federal, state, and local guidelines for laboratory animal care and are accredited by the Association for the Assessment and

Accreditation of Laboratory Animal Care International. De-identified human testis sections were obtained from Stanford University in compliance with protocol IRB-32801.

## Decision letter and Author response

Decision letter https://doi.org/10.7554/eLife.50209.sa1
Author response https://doi.org/10.7554/eLife.50209.sa2

## Additional files

### Supplementary files

- Supplementary file 1. Key Resources Table.

- Transparent reporting form

### Data availability

Microarray data has been deposited in Dryad, doi:10.5061/dryad.m262vd1; under the title "Clariom D microarray_HIPK4 null testes" Mass spectrometry data has been deposited in Dryad, doi:10.5061/dryad.m262vd1; under the title "MassSpec_HCD_NIH3T3-HIPK4_phospho-enriched" and "MassSpec_HCD_NIH3T3-HIPK4_total peptides".

The following dataset was generated:

| Author(s) | Year | Dataset title | Dataset URL | Database and Identifier |
|---|---|---|---|---|
| Crapster JA, Rack P, Hellmann Z, Elias J, Perrino J, Behr B, Li Y, Lin J, Zeng H, Chen J | 2019 | Data from: HIPK4 is essential for murine spermiogenesis | https://doi.org/10.5061/dryad.m262vd1 | Dryad Digital Repository, 10.5061/dryad.m262vd1 |

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
