## [Decision Letter]

**Acceptance summary:**

Within this publication, the authors have comprehensively defined the requirement for the kinase HIPK4 in spermatogenesis, and by extension male fertility. HIPK4 is required for the function of the acroplaxome, a poorly understood structure that sits at the interface between the sperm nucleus and the acrosome. This role is almost certainly achieved through the phosphorylation of key target proteins and the regulation of the actin cytoskeleton.

**Decision letter after peer review:**

Thank you for submitting your article "HIPK4 is essential for murine spermiogenesis" for consideration by *eLife*. Your article has been reviewed by two peer reviewers, including Moira K O'Bryan as the Reviewing Editor and Reviewer #1, and the evaluation has been overseen by Anna Akhmanova as the Senior Editor. The following individual involved in review of your submission has agreed to reveal their identity: Richard Oko (Reviewer #2).

The reviewers have discussed the reviews with one another and the Reviewing Editor has drafted this decision to help you prepare a revised submission.

Summary:

Within the publication the authors define homeodomain interacting protein kinase 4 (HIPK4) as a kinase with an essential role male fertility. *Hipk4* knockout males exhibit phenotypes consistent with oligoasthenoteratozoospermia in humans, and the sperm have reduced oocyte binding capacity in IVF, but produce viable offspring on ICSI. Based on ultrastructural and immunofluorescent analyses of spermiogenesis, showing drastic changes in head shape after step 8 in KO spermatids, combined with HIPK4 over-expression studies in cultured fibroblasts, where remodelling of the F-actin cytoskeleton occurs, the investigators conclude that the abnormal head shape phenotype was the likely cause of infertility, and that it was likely due to defects arising in the filamentous actin scaffolding in the acroplaxome during spermatid elongation. The paper is well written and the insights of value to the field. While extremely promising, the current data is not proof of the hypothesis that HIPK4 is a functionally relevant regulator of F-actin within the acroplaxome. Additional studies are required to achieve proof.

Essential revisions:

1) A detailed analysis of F-actin localization across the steps of spermiogenesis (both before and after step 8) is required. Notably, there is no direct proof of F-actin instability (or a phenotype) in the acroplaxome, or in the cytoplasm of spermatids, from steps 1 to 8, the time period corresponding to the expression of *Hipk4* mRNA. It was only after HIPK4 expression, during spermatid elongation and nuclear condensation that an overt phenotype became apparent. While it is possible the absence of phenotype is the result of a dependence on external (Sertoli) cell forces, no evidence, or discussion, is included in the manuscript. As such, we request a detailed analysis of F-actin across the relevant steps of spermiogenesis (ideally using both phalloidin and antibody methods). Localisation at an ultrastructural level would be best, however, we appreciate this is entirely dependent on the affinity of the antibody — please include a comment on this in the response to reviewers.

2) We also request an analysis/proof of the interaction between HIPK4 and actin within the acroplaxome. This could be achieved using a proximity ligation methods. We recommend using the knockout mouse as a control.

---

## [Author Response]

Essential revisions:1) A detailed analysis of F-actin localization across the steps of spermiogenesis (both before and after step 8) is required. Notably, there is no direct proof of F-actin instability (or a phenotype) in the acroplaxome, or in the cytoplasm of spermatids, from steps 1 to 8, the time period corresponding to the expression of Hipk4 mRNA. It was only after HIPK4 expression, during spermatid elongation and nuclear condensation that an overt phenotype became apparent. While it is possible the absence of phenotype is the result of a dependence on external (Sertoli) cell forces, no evidence, or discussion, is included in the manuscript. As such, we request a detailed analysis of F-actin across the relevant steps of spermiogenesis (ideally using both phalloidin and antibody methods). Localisation at an ultrastructural level would be best, however, we appreciate this is entirely dependent on the affinity of the antibody — please include a comment on this in the response to reviewers.

We appreciate the reviewers' concern that our original manuscript had no evidence of actin dysregulation in *Hipk4^â€“/â€“^* spermatids during the time period of HIPK4 expression, and we have investigated this question further by immunofluorescence microscopy. In addition to the fluorescently labeled phalloidin used for our previous studies, we have identified a commercially available anti-Î²-actin antibody that is an effective reagent for imaging the actin cytoskeleton in isolated spermatids. As suggested by the reviewers, we have employed these probes to analyze actin structures across the multiple steps of spermiogenesis, and our results are shown in Figure 7A (anti-Î²-actin antibody) and Figure 7â€”figure supplement 1 (Alexa Fluor 647-conjugated phalloidin). While phalloidin staining of isolated round spermatids was difficult to detect, possibly due to technical issues, we were able to achieve robust Î²-actin immunostaining at each step of spermatid differentiation. Using the anti-Î²-actin antibody, we can now observe actin defects in isolated *Hipk4^â€“/â€“^* spermatids occur at steps when HIPK4 would normally be expressed. Beginning at steps 5/6, approximately half of the isolated mutant spermatids had fragmented acrosomes, which were frequently uncoupled from the actin cytoskeleton. In comparison, all isolated wild-type round spermatids had intact acrosomes that remained closely associated with the underlying actin-scaffolded acroplaxome. We note that these cytoskeletal phenotypes become manifest earlier than those observed by transmission electron microscopy of testis sections, and we interpret this difference as further evidence for their dependence on external forces. Such forces are normally exerted by Sertoli cells during spermatid elongation, and it is likely that the process used to isolate round spermatids from dissociated testes applies similar forces to the acrosome–acroplaxome complex.

We also thank the reviewers for suggesting that we conduct ultrastructural studies of actin dynamics during spermiogenesis. While we have now found that confocal microscopy is sufficient to confirm HIPK4-dependent actin functions in found spermatids, we consulted with Prof. Pablo Visconti, who has recently used 3D-STORM to image F-actin in mature sperm (Gervasi, et al. J. Cell Sci. 2018, 131, jcs215897. doi:10.1242/jcs.215897). Dr. Visconti advised us against applying 3D-STORM to differentiating spermatids, as the experiments would be technically demanding and likely require several months to a year to complete. He instead suggested that we obtain super-resolution images of wild-type and *Hipk4^–/–^* spermatids by structured illumination microscopy (SIM). Unfortunately, those studies did not yield additional insights into how HIPK4 regulates the actin cytoskeleton (Author response image 1., and we have not pursued them further.

**Author response image 1. respfig1:** Two-dimensional structured illumination microscopy (SIM) of wild-type and *Hipk4^-/-^* spermatids. Spermatids were isolated from the testes of wild-type and *Hipk4^-/-^*mice, fixed, and then stained with a rabbit polyclonal anti-Î²-actin antibody and Alexa Fluor 488-conjugated anti-rabbit IgG secondary antibody to visualize the actin cytoskeleton. Acrosomal structures in the germ cells were stained with Alexa Fluor 647-conjugated peanut agglutinin (AF647-PNA). Scale bar: 3 Î¼m.

2) We also request an analysis/proof of the interaction between HIPK4 and actin within the acroplaxome. This could be achieved using a proximity ligation methods. We recommend using the knockout mouse as a control.

To investigate whether HIPK4 can interact with the actin cytoskeleton of spermatids, we isolated F-actin from whole testis lysates by ultracentrifugation. Strikingly, we observed that HIPK4 predominantly co-precipitates with the F-actin-containing pellet; only a small fraction of HIPK4 remains in the soluble fraction (which also contains G-actin). In addition, we found that loss of HIPK4 function specifically increases the amount of CAPZA2, an actin capping protein, that co-precipitates with F-actin. Together, these results suggest that HIPK4 exists primarily in complex with the actin cytoskeleton and/or actin-binding proteins. They also provide further evidence that HIPK4 regulates the actin cytoskeleton in spermatids.

**I**t is important to note that our mechanistic hypothesis does not require direct interactions between HIPK4 and the actin cytoskeleton within the acroplaxome. Our immunofluorescence studies demonstrate that HIPK4 protein is distributed throughout the cytoplasm of step 4-8 spermatids (the acrosomal staining in elongated spermatids is non-specific). We also determined that HIPK4 overexpression in NIH-3T3 cells alters the phosphorylation state of several proteins that can interact with the actin cytoskeleton but not that of actin subunits themselves. Based on these findings, we favor a model in which HIPK4 directly phosphorylates one or more regulatory proteins that control actin dynamics.

While we appreciate the reviewers' suggestion of proximity ligation assays, their propensity to yield false-positives is a complicating factor (bioRxiv 2018; doi: 10.1101/411355), especially since the direct substrate(s) of HIPK4 (and therefore appropriate specificity controls) remain unknown. We share the reviewers' interest in understanding how HIPK4 regulates actin within the acroplaxome, and this is the focus of our future studies.